# Neural Fields with Hard Constraints of Arbitrary Differential Order

**Fangcheng Zhong**
University of Cambridge

**Kyle Fogarty**
University of Cambridge

**Param Hanji**
University of Cambridge

**Tianhao Wu**
University of Cambridge

**Alejandro Sztrajman**
University of Cambridge

**Andrew Spielberg**
Harvard University

**Andrea Tagliasacchi**
Simon Fraser University

**Petra Bosilj**
University of Lincoln

**Cengiz Oztireli**
University of Cambridge

## Abstract

While deep learning techniques have become extremely popular for solving a broad range of optimization problems, methods to enforce hard constraints during optimization, particularly on deep neural networks, remain underdeveloped. Inspired by the rich literature on meshless interpolation and its extension to spectral collocation methods in scientific computing, we develop a series of approaches for enforcing hard constraints on neural fields, which we refer to as *Constrained Neural Fields* (CNF). The constraints can be specified as a linear operator applied to the neural field and its derivatives. We also design specific model representations and training strategies for problems where standard models may encounter difficulties, such as conditioning of the system, memory consumption, and capacity of the network when being constrained. Our approaches are demonstrated in a wide range of real-world applications. Additionally, we develop a framework that enables highly efficient model and constraint specification, which can be readily applied to any downstream task where hard constraints need to be explicitly satisfied during optimization. Source code is publicly available at `https://zfc946.github.io/CNF.github.io/`.

## 1 Introduction

Deep neural networks serve as universal function approximators [10] and have achieved tremendous success across a wide variety of tasks. Gradient-based learning algorithms enable optimizing the parameters of networks to approximate any desired model. However, despite the existence of numerous advanced optimization algorithms, the issue of enforcing strict equality constraints during training has not been sufficiently addressed. Constrained optimization has a broad spectrum of real-world applications. For example, in trajectory optimization ([38][Chapter 10], and, *e.g.*, [4, 29]), the agent poses at the start, end, and potentially intermediate steps are to be explicitly enforced. In signal representation, it is desirable for the continuous model to interpolate the discrete samples [22]. In physics, the models of interest must comply with fundamental physical principles [14, 27].

Applying traditional solvers for general constrained optimization, such as SQP [26], to neural networks can be nontrivial. Since traditional methods express constraints as a function of learnable parameters, this formulation becomes extremely high-dimensional, nonlinear, and non-convex in the context of neural networks. As the number of constraints increases, the computational complexity

37th Conference on Neural Information Processing Systems (NeurIPS 2023).

grows substantially. Other solutions for deep neural networks may encounter issues related to system conditioning, memory consumption, and network capacity when subject to constraints [20].

In this work, we consider the following constrained optimization problem:

$$\arg\min_{\theta} \mathcal{L}\left(f_\theta; \theta\right) \quad \text{s.t.} \quad \mathcal{F}\left[f_\theta\right]\left(\boldsymbol{x}\right) = g\left(\boldsymbol{x}\right) \; \forall \boldsymbol{x} \in \mathcal{S}, \tag{1}$$

where $f_\theta$ is a parametric model with learnable parameters $\theta$; and $\mathcal{F}$ is a linear operator[1] that transforms $f$ into another function $\mathcal{F}\left[f\right]$ which has the same input and output dimensions as $f$. Common examples of valid operators $\mathcal{F}$ include the identity map, integration, partial differentiation of arbitrary order, and many composite operators that involve differentiation, such as Laplacian, divergence, and advection operators. $\mathcal{F}$ can represent a broad class of constraints on the function $f$ in real-world applications. Enforcement of multiple constraints, expressed as different operators on $f$, is also possible. In practice, one typically does not have access to (or require) a closed-form expression of $g$, but rather the evaluation of $g$ on a set $\mathcal{S}$ of discrete samples of $\boldsymbol{x}$. In this work, we focus on equality constraints, as inequality constraints can be more easily satisfied through heavy regularization or a judicious choice of the activation function.

Our solution draws inspiration from meshless interpolation [6] and its extension to spectral collocation methods [7] in the scientific computing literature. These methods model a parametric function as a *linear sum of basis functions*. By computing the weights of the bases, hard constraints on the local behavior of the function at the *collocation points*, where the constraints need to be satisfied, are enforced accordingly. However, the selection of the basis functions also determines the inductive bias of the model, *i.e.*, the behavior of the function outside the collocation points. Our perspective is to formulate the problem of constrained optimization as a *collocation problem with learnable inductive bias*. Instead of using a set of fixed basis functions, we allow each basis function to have additional learnable parameters. While the weights of the basis functions are used to enforce constraints, other learnable parameters are optimized via a training loss to match the optimal inductive bias.

With this formulation of the problem, we demonstrate a series of approaches for enforcing hard constraints in the form of Eq. 1 onto a neural field, *i.e.*, a parametric function modeled as a neural network that takes continuous coordinates as input. In theory, we can formulate basis functions from existing neural fields of any architecture and enforce constraints. Nevertheless, certain network architectures may encounter difficulties as the number of collocation points or the order of differentiation increases. To tackle these challenges, we propose specific network designs and training strategies, and demonstrate their effectiveness in several real-world applications spanning physics, signal representation, and geometric processing. Furthermore, we establish a theoretical foundation that elucidates the superiority of our approach over existing solutions, and extend prior works with shared underlying designs to a broader context. Finally, we develop a PyTorch framework for the highly efficient adoption of Constrained Neural Fields (CNF), which can be applied to any downstream task requiring explicit satisfaction of hard constraints during optimization.

In summary, our work makes the following key contributions:

- We introduce a novel approach for enforcing linear operator constraints on neural fields;
- We propose a series of model representations and training strategies to address various challenges that may arise within the problem context, while offering a theoretical basis for CNF's superior performance and its generalization of prior works;
- Our framework is effective across a wide range of real-world problems; in particular, CNF:
    - achieves state-of-the-art performance in learning material appearance representations.
    - is the first that can enforce an exact normal constraint on a neural implicit field.
    - improves the performance of the Kansa method [18], a meshless partial differential equation (PDE) solver, when applied to irregular input grids.

## 2  Related Work

Applying constraints to a deep learning model has been a long-standing problem. Numerous studies have aimed to enforce specific properties on neural networks. For instance, sigmoid and softmax

---

[1]While our approach can be theoretically extended to nonlinear operators, we focus on linear ones due to the efficiency of linear solvers and GPU vectorization.

activation functions are used to ensure that the network output represents a valid probability mass function. Convolutional neural networks (CNNs) were introduced to address translation invariance. Pointnet [30] was proposed to achieve permutation invariance. POLICE [3] was introduced to enforce affine properties on the network. However, these works were not specifically designed to handle strict and general-form equality constraints. There are also studies that train a multilayer perceptron (MLP) [34, 37] or neural basis functions [8, 43] with a regression loss to approximate the equality constraints through overfitting, which we refer to as *soft constraint* approaches. However, in those works, strict adherence to hard constraints is rarely guaranteed. DC3 [12] is the first work that can enforce general equality constraints onto deep learning models by selecting a subset of network parameters for constraint enforcement and utilizing the remaining parameters for optimization. However, this approach cannot offer a theory or guidance regarding the existence of a solution when selecting a random subset of the network. In contrast, our method rigorously analyzes and guarantees the existence of a solution. Linearly constrained neural networks [17] can also explicitly satisfy linear operator constraints, but their method only applies to the degenerate case where $g(x)$ (Eq. 1) is zero everywhere. Neural kernel fields (NKF) [41] employs kernel basis functions similar to ours to solve constrained optimization problems in geometry, but their method cannot accommodate strict higher-order constraints. Their use of a dense matrix also cannot scale up to accommodate a large number of constraints, due to limitations in processor memory. A concurrent work, PDE-CL [24], utilizes a constrained layer to solve partial differential equations as a constrained optimization problem. We demonstrate that a constrained layer may not be the ideal choice for designing basis functions and propose alternative solutions. In fact, our approach can be seen as a *generalization* of NKF and PDE-CL. While seemingly unrelated, both works approach constrained optimization problems by leveraging the weights of basis functions to enforce constraints. They differ only in the design of the basis and the optimization function. Finally, there has been substantial development in implicit layers [13] that can enforce equality constraints between the layer's input and output while efficiently tracking gradients. However, this approach is not specifically designed for constraints over the entire network, although it can be potentially incorporated into our method to facilitate computation.

## 3   Method

At the core of our approach is the representation of a neural field $f\left(\cdot\right):\mathbb{R}^{M}\mapsto\mathbb{R}^{N}$ as a linear sum of basis functions, as with the collocation methods:

$$f\left(\boldsymbol{x}\right)=\sum_{i}\boldsymbol{\beta}_{i}\odot\Psi_{i}\left(\boldsymbol{x}\right),\tag{2}$$

where each $\Psi_{i}\left(\cdot\right):\mathbb{R}^{M}\mapsto\mathbb{R}^{N}$ represents a basis function that can be expressed as any parametric function with learnable parameters, such as neural networks; $\boldsymbol{\beta}_{i}\in\mathbb{R}^{N}$ is the weight of the $i$-th basis; and $\odot$ indicates the Hadamard product.

### 3.1   Constrained Optimization

Consider $\mathcal{S}:=\{\boldsymbol{x}_{i}\}_{i=1}^{I}$, a set of $I$ constraint points such that $\mathcal{F}\left[f\right]\left(\boldsymbol{x}_{i}\right)=g\left(\boldsymbol{x}_{i}\right),\forall i$ where the ground truth value of $g\left(\boldsymbol{x}_{i}\right)$ is accessible. Using the neural basis representation (2), and assuming that $\mathcal{F}$ is linear and $\mathcal{F}\left[\Psi_{i}\right]$ is well-defined for all $i$, the following needs to hold for the constraints:

$$\mathcal{F}\left[f\right]\left(\boldsymbol{x}\right)=\mathcal{F}\left[\sum_{i}\boldsymbol{\beta}_{i}\odot\Psi_{i}\right]\left(\mathbf{x}\right)=\sum_{i}\boldsymbol{\beta}_{i}\odot\mathcal{F}\left[\Psi_{i}\right]\left(\boldsymbol{x}\right)=g\left(\boldsymbol{x}\right),\ \forall\boldsymbol{x}\in\mathcal{S}.\tag{3}$$

Eq. 3 is a *collocation equation* in the context of spectral collocation methods for solving differential and integral equations [9, 1]. These constraints can be explicitly satisfied by defining $I$ basis functions and solving Eq. 3 for their weights. For a linear $\mathcal{F}$, we can expand Eq. 3 into batched matrix form:

$$\underbrace{\begin{bmatrix}\mathcal{F}\left[\Psi_{1}\right]\left(\boldsymbol{x}_{1}\right)&\mathcal{F}\left[\Psi_{2}\right]\left(\boldsymbol{x}_{1}\right)&\cdots&\mathcal{F}\left[\Psi_{I}\right]\left(\boldsymbol{x}_{1}\right)\\\mathcal{F}\left[\Psi_{1}\right]\left(\boldsymbol{x}_{2}\right)&\mathcal{F}\left[\Psi_{2}\right]\left(\boldsymbol{x}_{2}\right)&\cdots&\mathcal{F}\left[\Psi_{I}\right]\left(\boldsymbol{x}_{2}\right)\\\vdots&\vdots&\ddots&\vdots\\\mathcal{F}\left[\Psi_{1}\right]\left(\boldsymbol{x}_{I}\right)&\mathcal{F}\left[\Psi_{2}\right]\left(\boldsymbol{x}_{I}\right)&\cdots&\mathcal{F}\left[\Psi_{I}\right]\left(\boldsymbol{x}_{I}\right)\end{bmatrix}}_{\mathbf{A}_{f}}\begin{bmatrix}\boldsymbol{\beta}_{1}\\\boldsymbol{\beta}_{2}\\\vdots\\\boldsymbol{\beta}_{I}\end{bmatrix}=\begin{bmatrix}g\left(\boldsymbol{x}_{1}\right)\\g\left(\boldsymbol{x}_{2}\right)\\\vdots\\g\left(\boldsymbol{x}_{I}\right)\end{bmatrix}.\tag{4}$$

There are in total $N$ such matrix equations to solve, each corresponding to an output dimension of $f$ if $N>1^{2}$. We denote the matrix as $\mathbf{A}_{f}$ for reference. As long as we apply a differentiable linear

---

[2]In Einstein summation notation, the batched matrix equation in Eq. 4 can be expressed as $IIN, IN \rightarrow IN$.

| **Algorithm 1** Training | **Algorithm 2** Inference |
|---|---|
| 1: **repeat** | **Input:** $\boldsymbol{x}$ |
| 2:    Compute $\boldsymbol{\beta}$ as the solution of Eq. 4 | **Output:** $f_\theta(\boldsymbol{x})$ |
| 3:    Compute gradient $\frac{\partial \mathcal{L}}{\partial \theta} = \frac{\partial \mathcal{L}}{\partial f}\frac{\partial f}{\partial \theta}$ | 1: **if** $\boldsymbol{\beta}$ is **None then** |
| 4:    Update $\theta$ | 2:    Compute $\boldsymbol{\beta}$ from Eq. 4 w/o gradients |
| 5: **until** converged | 3: **end if** |
| | 4: $f_\theta(\boldsymbol{x}) \leftarrow \sum_i \boldsymbol{\beta}_i \odot \Psi_i(\boldsymbol{x})$ |

solver that provides valid gradients to compute the weights, the neural basis representation (Eq. 2) can be integrated with any optimization loss $\mathcal{L}(f_\theta; \theta)$ to approximate the desired behavior, while ensuring that the hard constraints are strictly enforced. Algorithm 1 describes the training procedure in detail, where Step 2 solves for the weights $\boldsymbol{\beta}$ in every iteration, ensuring consistent satisfaction of hard constraints throughout the *entire* training process, while Steps 3 and 4 update the training parameters $\theta$ to optimize for the training loss $\mathcal{L}$ under the constraints. We compute $\boldsymbol{\beta}$ using an LU decomposition with partial pivoting and row interchanges, which is fully differentiable when $\mathbf{A}_f$ is full rank. The computation of $\boldsymbol{\beta}$ constructs a computational graph that tracks $\frac{\partial \boldsymbol{\beta}}{\partial \theta}$, thereby ensuring a valid $\frac{\partial f}{\partial \theta}$. Consequently, any loss function with a valid gradient $\frac{\partial \mathcal{L}}{\partial f}$ can be smoothly integrated into this training procedure. The inference procedure is detailed in Algorithm 2. Here, $\boldsymbol{\beta}$ only needs to be pre-computed once since it is independent of the evaluation point $\boldsymbol{x}$. As a result, performing inference with CNF is highly efficient and boils down to computing a linear combination of $I$ basis functions without solving any matrix equations, a task that can be vectorized for efficiency.

This approach can also be extended to multiple linear operator constraints, indexed by $j$:

$$\mathcal{F}_j[f](\boldsymbol{x}) = g_j(\boldsymbol{x}), \forall \boldsymbol{x} \in \mathcal{S}_j, \tag{5}$$

where $S_j$ indicates the set of points $\boldsymbol{x}$ where the values of $\mathcal{F}_j[f](\boldsymbol{x})$ are constrained. Note that $S_j$ corresponding to different operator constraints can have repeated $\boldsymbol{x}$ elements. For example, to constrain $\nabla f(\boldsymbol{x})$ where $\boldsymbol{x} \in \mathcal{S} \subseteq \mathbb{R}^M$, a partial differentiation operator needs to be specified for each of the $M$ partial derivatives at the same point $\boldsymbol{x}$. These constraints can be explicitly satisfied by defining $\sum_j |S_j|$ basis functions and computing their weights by solving the batched matrix equation:

$$\begin{bmatrix} \mathcal{F}_1[\Psi_1](\boldsymbol{x}_1^1) & \mathcal{F}_1[\Psi_2](\boldsymbol{x}_1^1) & \cdots \\ \mathcal{F}_1[\Psi_1](\boldsymbol{x}_2^1) & \mathcal{F}_1[\Psi_2](\boldsymbol{x}_2^1) & \cdots \\ \vdots & \vdots & \ddots \\ \mathcal{F}_2[\Psi_1](\boldsymbol{x}_1^2) & \mathcal{F}_2[\Psi_2](\boldsymbol{x}_1^2) & \cdots \\ \mathcal{F}_2[\Psi_1](\boldsymbol{x}_2^2) & \mathcal{F}_2[\Psi_2](\boldsymbol{x}_2^2) & \cdots \\ \vdots & \vdots & \ddots \end{bmatrix} \begin{bmatrix} \boldsymbol{\beta}_1^1 \\ \boldsymbol{\beta}_2^1 \\ \vdots \\ \boldsymbol{\beta}_1^2 \\ \boldsymbol{\beta}_2^2 \\ \vdots \end{bmatrix} = \begin{bmatrix} g_1(\boldsymbol{x}_1^1) \\ g_1(\boldsymbol{x}_2^1) \\ \vdots \\ g_2(\boldsymbol{x}_1^2) \\ g_2(\boldsymbol{x}_2^2) \\ \vdots \end{bmatrix}, \tag{6}$$

where $\boldsymbol{x}_n^j$ indicates the $n$-th point in $\mathcal{S}_j$. The training and inference procedures are the same as previously described, except that $\boldsymbol{\beta}$ is now solved using Eq. 6 instead.

## 3.2 Neural Basis Functions

To construct a set of basis functions from an existing neural field, we consider a few approaches.

**Independent basis**    The most straightforward method is to specify a separate neural field for each basis. This method is the most expressive but can only be used with a small number of constraints. This is because the number of networks and learnable parameters increases linearly with the number of constraints. A bigger concern is the quadratic growth in the size of the matrix $\mathbf{A}_f$. As a result, even high-end processors are guaranteed to run out of memory working with a large number of constraints.

**Constraint basis**    We can directly use the last hidden layer of a multi-layer perceptron (MLP) as the basis and compute the weight of each node to enforce the constraints. This strategy has been referred to as a *constraint layer* in concurrent work [24]. The size of the network, excluding the hidden layer, does not grow as the number of constraints increases. However, this approach may lead to linearly-dependent basis functions, resulting in the matrix $\mathbf{A}_f$ being singular or having a *large condition number*. This can lead to a significant error bound in the satisfaction of hard constraints.

**Hypernetwork basis**  The third approach is to generate the parameters of each neural basis function using a hypernetwork [16]. The input to the hypernetwork is a one-hot encoding of length equal to the number of bases. The hypernetwork ensures that the network size does not increase with the number of basis functions. Nonetheless, the quadratic growth in the size of $\mathbf{A}_f$ remains an issue for both the constraint and the hypernetwork bases.

In several empirical studies, we found that the two aforementioned approaches hinder the network's capacities when subjected to hard constraints. They also encountered difficulties reducing the condition number; see Appendix.

**Kernel basis**  As a result, we propose several variants of a *neural kernel function* as a special design of basis functions:

$$\Psi_i(\boldsymbol{x}) = \kappa\left(\phi\left(\boldsymbol{x}_i\right), \phi\left(\boldsymbol{x}\right)\right), \tag{7}$$

where $\phi(\cdot)$ is a neural encoder that maps $\boldsymbol{x}$ into a high-dimensional feature space; $\kappa(\cdot, \cdot)$ is a kernel function that measures the similarity between two points in the feature space; $\boldsymbol{x}_i$ is the $i$-th constraint point, which we refer to as an *anchor point*. With this design, the higher-dimensional feature space bolsters the learning capacity of the basis function. The size of parameters in the network also does not grow with the number of constraints. If $\kappa(\cdot, \cdot)$ is a dot-product kernel, Eq. 7 becomes the same basis function used in NKF [41]. However, we empirically found that a Gaussian kernel better promotes the linear independence of basis functions due to its local compact support, while still preserving sufficient learning capacity; see Appendix.

**Hypernetwork kernel basis**  One drawback of the kernel basis function is that only one constraint can be applied to each anchor point. For example, one cannot enforce constraints on both the value of $f(\boldsymbol{x})$ and its gradient $\nabla f(\boldsymbol{x})$ at the same anchor point, as it would result in *repeated* basis functions $\Psi_i \equiv \Psi_j \equiv \kappa\left(\phi\left(\boldsymbol{x}_{\mathrm{anchor}}\right), \phi\left(\boldsymbol{x}\right)\right)$ and thus an ill-conditioned system; see Appendix for math details. For multiple constraints at repeated anchor points, we propose a *hypernet kernel function*:

$$\Psi_i(\boldsymbol{x}) = \kappa\left(\phi_i\left(\boldsymbol{x}_i\right), \phi_i\left(\boldsymbol{x}\right)\right), \tag{8}$$

where the weights of $\phi_i$ are controlled by a hypernetwork conditioned on the anchor point $\boldsymbol{x}_i$. This hypernet construction of the encoder ensures that the network would not have repeated basis functions and the number of network parameters does not grow with the number of basis functions.

**Hybrid kernel basis**  Central to our approach is the construction and inversion of a linear system, the size of which grows quadratically with the number of constraints. To mitigate severe memory issues in large-scale constraints, we design a basis that promotes sparsity in matrix $\mathbf{A}_f$ for large-scale problems. While using a Gaussian kernel can promote sparsity, it is difficult to select a proper bandwidth to explicitly control the number of nonzero elements since the kernel operates in the feature space rather than the original domain. To overcome this, we propose a *hybrid kernel function*:

$$\Psi_i(\boldsymbol{x}) = \kappa\left(\phi_i\left(\boldsymbol{x}_i\right), \phi_i\left(\boldsymbol{x}\right)\right) \kappa_G\left(\boldsymbol{x}_i, \boldsymbol{x}\right), \tag{9}$$

where the first part $\kappa\left(\phi_i\left(\mathbf{x}_i\right), \phi_i\left(\mathbf{x}\right)\right)$ can be either a Gaussian kernel basis or a hypernet kernel basis, depending on the task, and $\kappa_G$ is some compactly supported kernel function, allowing for explicit adjustment of the zero behavior of $\Psi_i$. A candidate $\kappa_G$ can be a *truncated* Gaussian kernel such as:

$$\kappa_G\left(\mathbf{x}_i, \mathbf{x}\right) = \begin{cases} \exp\left(-\frac{\|\mathbf{x}_i - \mathbf{x}\|^2}{2\sigma^2}\right) & \text{if } \|\mathbf{x}_i - \mathbf{x}\| < 3\sigma \\ 0 & \text{if } \|\mathbf{x}_i - \mathbf{x}\| \geq 3\sigma. \end{cases} \tag{10}$$

Reducing the magnitude of $\sigma$ yields a matrix $\mathbf{A}_f$ with more zero entries, breaking the quadratic dependency on the number of constraint points.

Properties of various basis functions are summarized in Tab. 1. We recommend using the regular Gaussian kernel basis for standard-scale problems without multiple constraints at repeated anchor points, and employing hypernet kernel and hybrid kernel for corresponding advanced scenarios. In Sec. 4, we demonstrate the application of each type of basis function in concrete real-world examples.

### 3.3  Training Strategies

**Regularization**  In many applications, we recommend incorporating an additional loss term to regularize the condition number of the matrix $\mathbf{A}_f$ during optimization. A low condition number corresponds to a reduced level of matrix singularity and a smaller error bound in the solution vector, which is crucial for satisfying hard constraints.

Table 1: Basis functions summary.

| | Independent basis | Constraint basis | Hypernetwork basis | Kernel basis (Dot-product) | Kernel basis (Gassian) | Hypernetwork kernel basis (Gassian) | Hybrid kernel basis (Gassian) |
|---|---|---|---|---|---|---|---|
| Learning capacity | | Poor | Poor | Fair | Good | Good | Good |
| Linear independence | | Poor | Fair | Poor | Good | Good | Good |
| Matrix sparsity | Dense | Dense | Dense | Dense | Sparse | Sparse | Sparse |
| Params # independent of constraints # | No | Yes | Yes | Yes | Yes | Yes | Yes |
| Controllable sparsity | No | No | No | No | No | No | Yes |
| Multiple constraints at repeated anchor points | Yes | Yes | Yes | No | No | Yes | Yes (if using hypernet) |

**Transfer learning** Since CNF is formulated as a collocation method with a learnable inductive bias, we can pre-train the inductive bias of basis functions with custom configurations, such as well-conditioning, smoothness, or more advanced variations. This allows us to apply the pre-trained basis functions to unseen data, directly computing the weights with little or no additional training, as the problem reduces to a near-collocation scenario. We demonstrate this by solving a partial differential equation on an irregular grid in *inference time* and *without* further training; see Appendix.

**Sparse solver** Drawing inspiration from classical reconstruction techniques [23, 40], we introduce a *patch-based* approach that leverages the restricted support induced by the hybrid kernel function (Eq. 9) to solve the sparse linear system. For each evaluation point $x$, we construct a subset of basis functions such that $\Psi_i(x) \neq 0$ using the support criterion induced by $\kappa_G$. This enables us to create a lower-dimensional dense submatrix $(\mathbf{A}_f)_{\kappa_G}$ to solve for the weights of only the nonzero items so that the size of the system is sufficiently small to fit within the processor's memory limitations.

## 4 Applications

For efficient adoption of CNF in real-world applications, we have developed a PyTorch framework that abstracts the definition of the basis function, enabling users to conveniently define and experiment with custom architectures. The framework also enables vectorized basis function operations for highly efficient training and inference. Additionally, users have the flexibility to specify an arbitrary linear differential operator constraint by providing a regular expression in LaTeX. This can be processed to compute the exact post-operation function evaluation via automatic differentiation, thereby eliminating the need for manual execution. Further clarification of the features and usage of our framework, along with illustrative examples, is provided alongside the code release.

With this framework, we leverage CNF in four real-world applications, utilizing distinct basis functions tailored to the specific context of the problem. We elucidate and empirically demonstrate the merits of the chosen basis function in each scenario. Our intention is to use these examples as a guideline for selecting appropriate basis functions, which has been briefly summarized at the end of Sec. 3.2. Apart from Sec. 4.1, which serves as a toy example, we demonstrate unique advantages or state-of-the-art results achieved in each application when employing CNF.

### 4.1 Fermat's Principle

Fermat's principle in optics states that the path taken by a ray between two given points is the path that can be traveled in the least time. This can be formulated as a constrained optimization problem:

$$\arg\min_{\mathcal{C}} \int_{\mathcal{C}} \frac{d\mathbf{s}}{v} \quad \text{s.t.} \quad \partial\mathcal{C} = \{\mathbf{p}_0, \mathbf{p}_1\}, \tag{11}$$

where $d\mathbf{s}$ is the differential arc length that the ray travels along contour $\mathcal{C}$ and $v$ is the light speed that varies along the contour due to the refractive index of the medium. The contour's entry and exit points $\partial\mathcal{C}$ are hard constraints and the contour is optimized to minimize the travel time $\int_{\mathcal{C}} \frac{d\mathbf{s}}{v}$. We model the contour $\mathcal{C}$ as a collection of points from a parametric equation, i.e. $\mathcal{C} := \{\mathbf{p}(x) := \beta_1 f_1(x) + \beta_2 f_2(x) \mid x \in [0, 1]\}$. As there are in total two constraint points, we use two independent quadratic polynomials $\{f_1, f_2\}$ as basis functions. We chose polynomials over neural networks due

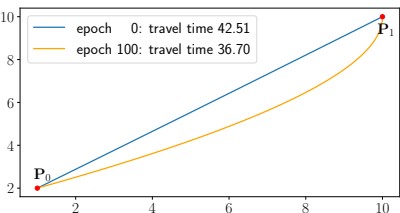

Figure 1: The optical paths and travel time before and after the optimization. The constraints are strictly enforced throughout the optimization.

Table 2: Quantitative evaluation of learning material appearance representations. We evaluate each method by rendering the learned BRDFs with environment map illumination and computing the mean and standard deviation of image metrics across all materials in the MERL dataset [21].

| | RMSE$_{\times 10^{-2}}\downarrow$ | PSNR$\uparrow$ | SSIM$\uparrow$ |
|---|---|---|---|
| NBRDF [35] | $1.57 \pm 2.86$ | $44.96 \pm 11.55$ | $.96 \pm .07$ |
| FFN [37] | $1.16 \pm 1.36$ | $44.72 \pm 10.46$ | $.98 \pm .03$ |
| SIREN [34] | $3.32 \pm 3.05$ | $33.43 \pm 8.84$ | $.93 \pm .08$ |
| Kernel FFN | $0.94 \pm 1.16$ | $46.32 \pm 9.92$ | $.98 \pm .03$ |
| Kernel SIREN | $\mathbf{0.60 \pm 0.54}$ | $\mathbf{47.95 \pm 7.99}$ | $\mathbf{.99 \pm .01}$ |

to the simplicity of the problem. The weights $\{\beta_1, \beta_2\}$ are solved differentially to fit hard constraints, while the coefficients of the polynomials are learnable parameters that can be updated with respect to the loss function. We show the results in Fig. 1 with an increasing refractive index along the Y-axis.

## 4.2 Learning Material Appearance

In visual computing, the synthesis of realistic renderings fundamentally requires an accurate description of the reflection of light on surfaces [11]. This is commonly expressed by the Bidirectional Reflectance Distribution Function (BRDF), described by Nicodemus *et al.* [25], which quantifies the ratio of incident and outgoing light intensities for a single material: $f_r(\boldsymbol{\omega}_i, \boldsymbol{\omega}_o)$, where $\boldsymbol{\omega}_i, \boldsymbol{\omega}_o \in S^2$ are the incident and outgoing light directions.

Our aim is to learn a neural field $\Phi_\theta(\boldsymbol{\omega}_i, \boldsymbol{\omega}_o)$ with parameters $\theta$ that closely matches the ground truth BRDF $f_r(\boldsymbol{\omega}_i, \boldsymbol{\omega}_o), \forall \boldsymbol{\omega}_i, \boldsymbol{\omega}_o \in \mathcal{S}^2$. We constrain $\Phi_\theta$ by selecting 100 sample pairs of $(\hat{\boldsymbol{\omega}}_i, \hat{\boldsymbol{\omega}}_o)$, where half of the pairs are uniformly sampled, and the other half of $\hat{\boldsymbol{\omega}}_o$ are concentrated around the direction of the perfect reflection of $\hat{\boldsymbol{\omega}}_i$. This means that the latter constraint points are situated around the specular highlight, which contains high-frequency details that regular neural networks struggle to learn [32, 31, 19, 35]. We illustrate these issues in Fig. 2, where we compare CNF, formulated with a Gaussian kernel basis (Eq. 7), with SIREN [34] or FFN [37] as the encoder network, against multiple state-of-the-art neural-based fitting approaches [34, 37, 35] for BRDF reconstruction. We perform this evaluation on highly specular materials from the MERL dataset [21], with size-matching networks for a fair assessment, and subsequently rendering a standard fixed scene; see Appendix for the complete MERL rendering results. Following previous work [35], we train $\Phi_\theta$ on 640k $(\boldsymbol{\omega}_i, \boldsymbol{\omega}_o)$ samples, minimizing L1 loss in the logarithmic domain to account for the large variation of BRDF values due to the specular highlight, while enforcing the aforementioned hard constraints on $(\hat{\boldsymbol{\omega}}_i, \hat{\boldsymbol{\omega}}_o)$:

$$\arg \min_{\theta} \mathbb{E}_{\boldsymbol{\omega}_i, \boldsymbol{\omega}_o} |\Phi_\theta(\boldsymbol{\omega}_i, \boldsymbol{\omega}_o) - \log\left(f_r(\boldsymbol{\omega}_i, \boldsymbol{\omega}_o) + 1\right)|$$
$$\text{s.t. } \Phi_\theta(\hat{\boldsymbol{\omega}}_i, \hat{\boldsymbol{\omega}}_o) = \log\left(f_r(\hat{\boldsymbol{\omega}}_i, \hat{\boldsymbol{\omega}}_o) + 1\right). \tag{12}$$

We summarize the results computed over all MERL materials in Tab. 2, where CNF demonstrates superior performance in learning material appearances with state-of-the-art accuracy; see Appendix for additional evaluations and ablation studies.

## 4.3 Interpolatory Shape Reconstruction

The representation of shapes via neural implicit functions [15] has recently garnered much attention [28, 2, 34]. Compared to traditional forms of representation such as point clouds, meshes, or voxel grids, neural implicit functions offer a flexible and continuous representation capable of handling unrestricted topology. In this section, we demonstrate how CNF can be applied to enforce exact interpolation constraints on implicit shape representations, while retaining the ability to train the field with geometry-specific inductive bias. Given an *oriented* point cloud $\mathcal{P} := \{(\boldsymbol{x}_i, \boldsymbol{n}_i)\}_{i \in I}$, our aim is to construct a continuous implicit function $\Phi(\boldsymbol{x})$ on the domain $\Omega$ whose 0-level-set $\mathcal{S} := \{\boldsymbol{p} \in \Omega \mid \Phi(\boldsymbol{p}) = 0\}$ represents the surface of the geometry.

**Exact normal constraints** We first demonstrate how our framework can be utilized to learn an implicit shape representation, while exactly interpolating both surface points and their associated

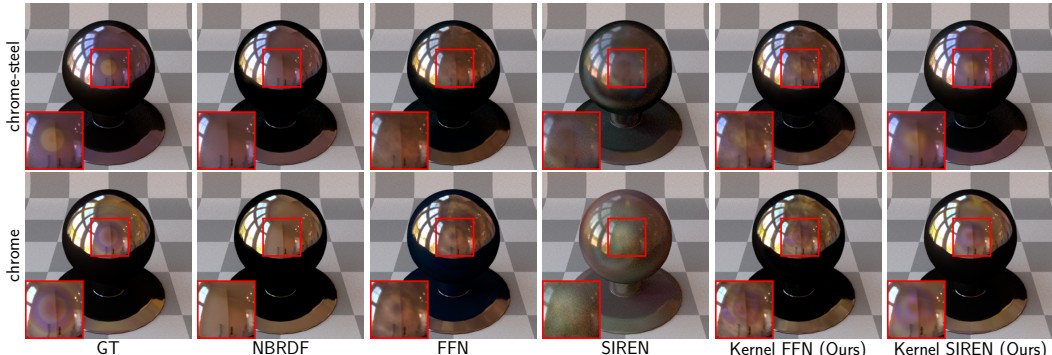

Figure 2: Learned material appearance by enforcing hard constraints around the specular highlights.

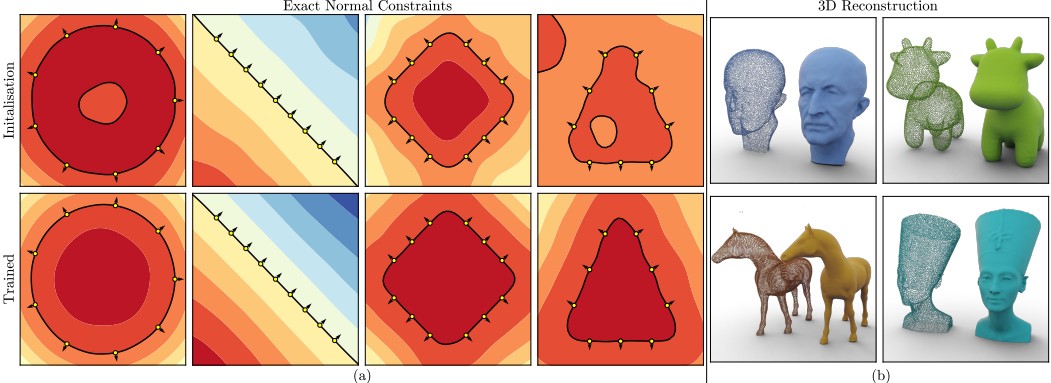

Figure 3: (a): Learning shapes from 2D point clouds (*yellow points*) with oriented normals (*black arrows*) using CNF; black lines depict the zero level sets of the learned implicit field. The method interpolates both exact point and normal constraints during initialization and throughout training, yielding a plausible surface upon minimizing the Eikonal constraint across the field. (b): Surface reconstruction of 3D point clouds containing 10,000 points.

normals. We conduct experiments in 2D, and formulate CNF with a hypernetwork kernel basis (Eq. 8), as the problem involves both zero-order and higher-order constraints at repeated constraint points. Note that this approach is the *first* to enforce exact normal constraints on a neural implicit field, as prior works [34, 41] can only approximate the exact normal with a pseudo-normal constraint [39]. In contrast, we constrain the implicit field such that $\Phi(\boldsymbol{x}_i) = 0, \forall \boldsymbol{x}_i \in \mathcal{P}$ and $\nabla\Phi(\boldsymbol{x}_i) = \boldsymbol{n}_i, \ \forall \boldsymbol{x}_i, \boldsymbol{n}_i \in \mathcal{P}$. Inspired by the geometrically motivated initialization [2], we use weights that are pre-trained to represent the signed distance function (SDF) of a circle with the desired number of constraint points. We found this placed the network in a more favorable starting position to produce plausible surfaces under new constraint points. We then train the network to solve the Eikonal equation, following previous work [15], via minimizing the quadratic loss function:

$$\mathcal{L} = \mathbb{E}_{\boldsymbol{x}} \left( \|\nabla_{\boldsymbol{x}}\Phi(\boldsymbol{x})\| - 1 \right)^2, \tag{13}$$

where the expectation is taken with respect to a uniform distribution over the domain $\Omega$. As shown in Fig. 3(a), CNF produces plausible surfaces to explain the point cloud, with the advantage of *not* depending on manufactured offset points [39] to ensure the existence of a non-trivial implicit surface.

**Large-scale constraints**  To demonstrate our hybrid kernel approach (Eq. 9) with the sparse solver for handling large-scale constraints, we reconstruct 3D point clouds comprising 10,000 points uniformly sampled over the surface. We set the support radius of the kernel to four times the average nearest neighbor distance, and assign a constant value of $10^5$ to regions of space with no support. The reconstruction is obtained by adhering to the points and pseudo-normal constraints [39, 41] during initialization, with no additional training required for the dense reconstruction to produce smooth surfaces, as shown in Fig. 3(b); see Appendix for additional results and evaluations.

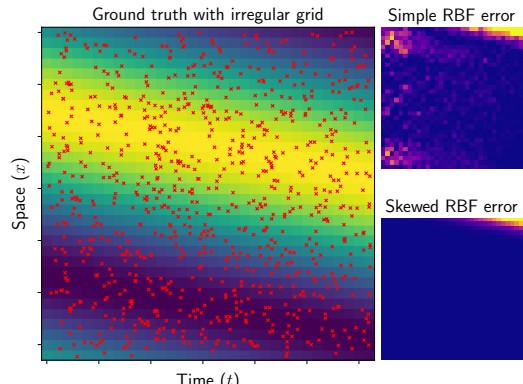

**Table 3:** Solving the 1D advection PDE ($\beta = 0.1$) with a simple (left) and skewed (right) RBF kernel. Both versions were optimized by our framework. We can see the benefit of using the skewed RBF at various perturbation levels.

| Pertubation | RBF | | Skewed RBF | |
|---|---|---|---|---|
| | RMSE | nRMSE | RMSE | nRMSE |
| 0.01 | 0.0086 | 0.0149 | **0.0061** | **0.0116** |
| 0.05 | 0.0252 | 0.0442 | **0.0024** | **0.0043** |
| 0.10 | 0.1484 | 0.2534 | **0.0344** | **0.0651** |
| 0.50 | 0.0170 | 0.0297 | **0.0167** | **0.0282** |
| 1.00 | 0.2830 | 0.4838 | **0.0231** | **0.0365** |

Figure 4: Left: Ground truth advection function and skewed input grid with $\sigma = 0.1$ perturbation. Right: Error maps for simple and skewed RBF.

## 4.4 Self-tuning PDE Solver

Our framework can also be applied to solve partial differential equations (PDEs). If using fixed radial basis functions (RBFs) without optimization, CNF simplifies to the *Kansa method* [18], a type of spectral collocation method where partial derivatives are enforced via hard constraints. Compared to the classical mesh-based numerical PDE solvers such as the finite difference method (FDM) and finite element method (FEM), Kansa has the advantage of offering an analytical and closed-form solution function across the entire domain, and solving a PDE on an *irregular* grid without requiring interpolation or meshing, *i.e.* grouping the points into triangles or quadrilaterals. Meshing on an irregular grid proves to be extremely difficult and severely harms the accuracy of these methods [9]. However, one disadvantage of the Kansa method is the difficulty in choosing the appropriate RBF kernel [5]. When using a Gaussian kernel, a large bandwidth may lead to an ill-conditioned system, while a small bandwidth may break the global smoothness of the solution function. This is especially challenging when the number or dimensionality of collocation points is large, or when the points are closely or irregularly spaced, making it difficult to manually fine-tune the hyperparameters of the RBF. Our method addresses this limitation by supporting the automatic fine-tuning of hyperparameters, optimizing both the conditioning and global smoothness of the solution:

$$\arg \min_{\theta} \ \omega_1 \operatorname{cond}(\mathbf{A}_{f_\theta}) + \omega_2 \underbrace{\int |\nabla f_\theta(\boldsymbol{x})| \, d\boldsymbol{x}}_{\text{total variation}} \quad \text{s.t.} \quad \mathcal{F}_j [f] (\boldsymbol{x}) = g_j (\boldsymbol{x}) \ \forall j, \qquad (14)$$

where $\operatorname{cond}(\mathbf{A}_f)$ is the condition number of the linear system and $\mathcal{F}_j$ corresponds to the PDE constraints along with the initial and boundary conditions. We demonstrate this approach by solving the following 1D advection equation [36] on an irregular grid:

$$\frac{\partial u(x,t)}{\partial t} + \beta(x) \frac{\partial u(x,t)}{\partial x} = 0, \qquad x \in (0,1), \ t \in (0,1),$$
$$u(x, t = 0) = u_0(x), \qquad x \in (0,1), \qquad\qquad\qquad (15)$$

where $u_0(x)$ represents the initial state of the system. We sample 32 points on the boundary $t = 0$ and $31 \times 32$ points on the space-time domain $(0,1) \times (0,1)$, which amount to a total of 1024 points, as hard constraints. The non-boundary samples, shown as red crosses in Fig. 4 are generated by a uniform grid perturbed by Gaussian noise. In this example, we cannot apply a single unified RBF bandwidth to fit such an irregular grid. Rather, we design a *skewed radial basis function*:

$$\kappa'_i (\boldsymbol{x}_i, \boldsymbol{x}) = \exp \left( -\frac{1}{2}(\boldsymbol{x} - \boldsymbol{x}_i)^T \boldsymbol{\Sigma}_i^{-1}(\boldsymbol{x} - \boldsymbol{x}_i) \right), \qquad (16)$$

where $\boldsymbol{x} = (x, t)$ and $\boldsymbol{\Sigma}_i = \operatorname{diag}([a_i, b_i])$ is a per-basis learnable covariance matrix. Our results in Tab. 3 show that using the skewed RBF kernel and optimizing per-basis parameters consistently improves performance on the irregular grid; see Appendix for additional results and evaluations.

The skewed RBF is essentially another variant of the Gaussian kernel (Eq. 7) but without the neural encoder. We recommend using neural networks only when the behavior of the basis functions to be optimized away from the constraint points is highly complex, as seen in the BRDF and shape-reconstruction examples. For problems where the desirable priors are as simple as linear independence and smoothness, such as solving PDEs, our proposed skewed RBF demonstrates sufficient capacity.

## 5 Discussion

**Training and inference efficiency** CNF is highly efficient in training (minutes) and inference (seconds); see Appendix for detailed reports, learning curves, and additional evaluations.

**Theoretical analysis** We offer a theoretical basis for CNF's superior performance:

*Comparison with general solvers* Most numerical methods for constrained optimization in scientific computing, including the popular SQP [26] and the more recent DC3 [12], adopt the following formulation for the constrained optimization problem:

$$\arg\min_{\theta} \mathcal{L}(\theta) \quad \text{s.t.} \quad h_1(\theta) = 0, \ h_2(\theta) = 0, \ ... \tag{17}$$

where $\theta$ denotes the learnable parameters. For problems where $\theta$ represents the weights of a deep neural network, these constraints become extremely high-dimensional and nonlinear, especially as the number of constraints increases. In this formulation, a constraint is only linear when $h(\theta)$ is a linear function of $\theta$. Therefore, linearity no longer holds as $h(\theta)$ involves constraints of a neural function. In contrast, our formulation (Eq. 1, 3, and 5) only requires the operator $\mathcal{F}$ to be linear, while the neural function $f$ can still be highly nonlinear with respect to $\theta$. Hence, CNF significantly reduces the problem's complexity, enabling the explicit determination and promotion of a solution's existence.

*Comparison with soft constraint approaches* While there have been attempts to model constraints by overfitting a neural function trained with regression [34, 37], CNF offers several clear advantages over such soft approaches. Most notably, CNF satisfies hard constraints without the need for training, whereas soft approaches may require extensive training to approximate constraints. Furthermore, despite extensive training, soft approaches may still fail to satisfy hard constraints due to inherent limitations in their learning capacity. In contrast, CNF provides a robust guarantee of hard constraint satisfaction within machine precision error, provided the condition number is small. Another drawback of employing soft approaches becomes evident when effectively imposing priors. The incorporation of priors often involves introducing additional terms to the loss. Consequently, a tradeoff arises between the constraints and the prior, which is controlled by a hyperparameter. In contrast, CNF offers a clean solution without such a tradeoff. With CNF, the inclusion of priors does not compromise hard constraints, thereby maintaining a harmonious balance among various aspects of the model.

*Generalization of NKF and PDE-CL* CNF generalizes prior works NKF [41] and PDE-CL [24]. NKF employed a dot-product kernel basis for 3D reconstruction, while PDE-CL used a constraint basis to solve PDEs. Yet, these bases exhibit suboptimal performance in terms of linear independence and learning capacity. Their use of dense matrices also renders them impractical for large-scale problems. Moreover, the dot-product kernel used by NKF cannot address strict higher-order constraints; see Appendix. We propose several novel variants of basis functions to enhance linear independence, learning capacity, and matrix sparsity, with strategies for analyzing and facilitating convergence. Our methodology unifies NKF and PDE-CL, extending their formulations to a broader theoretical context.

**Extension to nonlinear operator constraints** Although we restrict the operator to be linear in this work, CNF can be extended to accommodate nonlinear operators by solving a nonlinear version of Eq. 3, provided that the solver is differentiable. This could be accomplished through the use of implicit layers [13], which we leave as future work.

## 6 Summary

We introduce Constrained Neural Fields (CNF), a method that imposes linear operator constraints on deep neural networks, and demonstrate its effectiveness through real-world examples. As constrained optimization is a fundamental problem across various domains, we anticipate that CNF will unlock a host of intriguing applications in fields such as physics, engineering, finance, and visual computing.

## Acknowledgments and Disclosure of Funding

This work was supported by the UKRI Future Leaders Fellowship [G104084] and the Engineering and Physical Sciences Research Council [EP/S023917/1].

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

# Neural Fields with Hard Constraints
# of Arbitrary Differential Order

# Appendix

In this appendix, we provide additional evaluations of various designs of neural basis functions (Sec. A), along with additional details and results regarding their applications in material appearance learning (Sec. B), interpolatory shape reconstruction (Sec. C), and the self-tuning PDE solver (Sec. D), which also includes a *transfer learning* example (Sec. D.3).

## A  Neural Basis Functions

### A.1  Issues with the Neural Kernel Basis Function

We illustrate how, when using the neural kernel basis function (Eq. 7), enforcing multiple linear operator constraints at the same anchor point would result in *repeated* basis functions $\Psi_i \equiv \Psi_j \equiv \kappa\left(\phi\left(\boldsymbol{x}_{\text{anchor}}\right), \phi\left(\boldsymbol{x}\right)\right)$, consequently leading to an *ill-conditioned* linear system. Recall Eq. 6:

$$
\begin{bmatrix}
\mathcal{F}_1\left[\Psi_1\right]\left(\boldsymbol{x}_1^1\right) & \mathcal{F}_1\left[\Psi_2\right]\left(\boldsymbol{x}_1^1\right) & \cdots \\
\mathcal{F}_1\left[\Psi_1\right]\left(\boldsymbol{x}_2^1\right) & \mathcal{F}_1\left[\Psi_2\right]\left(\boldsymbol{x}_2^1\right) & \cdots \\
\vdots & \vdots & \ddots \\
\mathcal{F}_2\left[\Psi_1\right]\left(\boldsymbol{x}_1^2\right) & \mathcal{F}_2\left[\Psi_2\right]\left(\boldsymbol{x}_1^2\right) & \cdots \\
\mathcal{F}_2\left[\Psi_1\right]\left(\boldsymbol{x}_2^2\right) & \mathcal{F}_2\left[\Psi_2\right]\left(\boldsymbol{x}_2^2\right) & \cdots \\
\vdots & \vdots & \ddots
\end{bmatrix}
\begin{bmatrix}
\boldsymbol{\beta}_1^1 \\
\boldsymbol{\beta}_2^1 \\
\vdots \\
\boldsymbol{\beta}_1^2 \\
\boldsymbol{\beta}_2^2 \\
\vdots
\end{bmatrix}
=
\begin{bmatrix}
g_1(\boldsymbol{x}_1^1) \\
g_1(\boldsymbol{x}_2^1) \\
\vdots \\
g_2(\boldsymbol{x}_1^2) \\
g_2(\boldsymbol{x}_2^2) \\
\vdots
\end{bmatrix},
\tag{18}
$$

where $\boldsymbol{x}_n^j$ indicates the $n$-th point in $\mathcal{S}_j$ such that $\mathcal{F}_j\left[f\right]\left(\boldsymbol{x}\right) = g_j\left(\boldsymbol{x}\right), \forall\, \boldsymbol{x} \in \mathcal{S}_j$. When applying the neural kernel basis function $\Psi_i\left(\boldsymbol{x}\right) = \kappa\left(\phi\left(\boldsymbol{x}_i\right), \phi\left(\boldsymbol{x}\right)\right) = \kappa_N\left(\boldsymbol{x}_i, \boldsymbol{x}\right)$, the above matrix equation becomes:

$$
\begin{bmatrix}
\mathcal{F}_1\kappa_N(\boldsymbol{x}_1^1, \boldsymbol{x}_1^1) & \mathcal{F}_1\kappa_N(\boldsymbol{x}_2^1, \boldsymbol{x}_1^1) & \cdots \\
\mathcal{F}_1\kappa_N(\boldsymbol{x}_1^1, \boldsymbol{x}_2^1) & \mathcal{F}_1\kappa_N(\boldsymbol{x}_2^1, \boldsymbol{x}_2^1) & \cdots \\
\vdots & \vdots & \ddots \\
\mathcal{F}_2\kappa_N(\boldsymbol{x}_1^1, \boldsymbol{x}_1^2) & \mathcal{F}_2\kappa_N(\boldsymbol{x}_2^1, \boldsymbol{x}_1^2) & \cdots \\
\mathcal{F}_2\kappa_N(\boldsymbol{x}_1^1, \boldsymbol{x}_2^2) & \mathcal{F}_2\kappa_N(\boldsymbol{x}_2^1, \boldsymbol{x}_2^2) & \cdots \\
\vdots & \vdots & \ddots
\end{bmatrix}
\begin{bmatrix}
\boldsymbol{\beta}_1^1 \\
\boldsymbol{\beta}_2^1 \\
\vdots \\
\boldsymbol{\beta}_1^2 \\
\boldsymbol{\beta}_2^2 \\
\vdots
\end{bmatrix}
=
\begin{bmatrix}
g_1(\boldsymbol{x}_1^1) \\
g_1(\boldsymbol{x}_2^1) \\
\vdots \\
g_2(\boldsymbol{x}_1^2) \\
g_2(\boldsymbol{x}_2^2) \\
\vdots
\end{bmatrix},
\tag{19}
$$

Without loss of generality, assuming $f : \mathbb{R} \mapsto \mathbb{R}$; $\mathcal{F}_1$ is identity map; $\mathcal{F}_2$ is differentiation; and $\mathcal{S}_1 = \mathcal{S}_2 = \{x_0\}$ (same anchor point for both constraints), the matrix equation reduces to:

$$
\underbrace{\begin{bmatrix}
\kappa_N(x_0, x_0) & \kappa_N(x_0, x_0) \\
\kappa_N'(x_0, x_0) & \kappa_N'(x_0, x_0)
\end{bmatrix}}_{\text{singular}}
\begin{bmatrix}
\beta_1 \\
\beta_2
\end{bmatrix}
=
\begin{bmatrix}
g_1(x_0) \\
g_2(x_0)
\end{bmatrix}.
\tag{20}
$$

Therefore, the system is ill-conditioned as the matrix has repeated columns. NKF [41] uses this basis function with a dot-product kernel and thus cannot enforce strict constraints simultaneously on both the points and their normals in shape reconstruction, which we further evaluate in Sec. C.1.1. We also discovered significant disadvantages of using a dot-product kernel as opposed to a Gaussian kernel. We elaborate on these findings in Sec. A.2 and Sec. B.2.

### A.2  Basis Function Ablation

Choosing the appropriate basis function for an application requires consideration of two fundamental properties of basis functions: their linear independence and learning capacity. Linear independence is crucial for satisfying hard constraints, whereas learning capacity is key for fitting the inductive bias. In this ablation study, we train various neural basis functions discussed in Sec. 3.2 to regularize the condition number of the matrix $\mathbf{A}_f$, which is used to constrain a 2D function over 4096 points. A large condition number often corresponds to a substantial error in satisfying hard constraints. This study offers insights into how each design is prone to the ill-conditioning of the linear system. For a fair comparison, we use the same architecture for all neural encoders—a 2-layer MLP with Tanh activations and a total of 1.3M trainable parameters.

As shown in Fig. 5, the Gaussian kernel basis is well-conditioned throughout training, while all other methods struggle to reduce the condition number.

In Sec. B.2, we also discovered that the Gaussian kernel basis, when constrained, exhibits the superior learning capacity for fitting the BRDF. In contrast, the performance of other methods was found to be extremely poor. Note that the dot-product kernel and the constrained layer are the basis functions used in [41] and [24], respectively. Based on our findings, we strongly recommend employing the Gaussian kernel basis function when performing constrained optimization for neural fields.

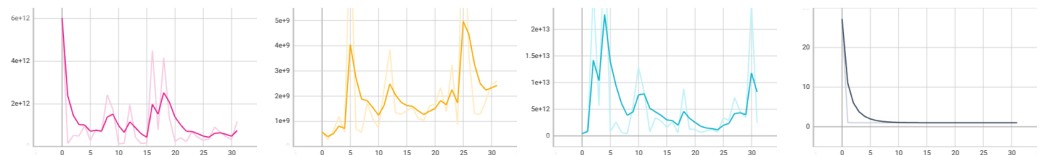

Figure 5: Condition number of different neural basis functions in the following order (left to right): constraint layer (pink), hypernet basis (yellow), dot-product kernel (blue), and Gaussian kernel (black) throughout training.

# B    Learning Material Appearance

## B.1    Experiment Details

In the experiments on material appearance fitting, all neural networks have a single hidden layer and we controlled the network width so that all methods have roughly 200k parameters for a fair comparison. Both baseline FFN and kernel FFN use a frequency of 16, meaning that the inputs are mapped to encodings of size 38 before feeding to the MLP. The kernel FFN model uses FFN as an encoder which has a width of 248 and outputs a latent vector of dim 512. The baseline FFN MLP has a width of 718. The kernel SIREN model has a width of 256 and also outputs a latent vector of size 512. The baseline SIREN and NBRDF have a width of 442. We use Adam optimizer with a learning rate of $5 \times 10^{-4}$ for all methods except for SIREN baseline and kernel SIREN, which use a learning rate of $1 \times 10^{-4}$ for more stable performance.

For constraint points, we sample the angles $\theta_h$, $\theta_d$ and $\phi_d$ as in Rusinkiewicz's parameterization [33]. A figure of those angles is shown in Fig. 6. We sample $\theta_d$ and $\phi_d$ uniformly at random within range $[0, \pi/2]$ and $[0, 2\pi]$ respectively. Half of the $\theta_h$ are also sampled uniformly at random within $[0, \pi/2]$, whereas the other half are sampled from a Gaussian distribution with a mean of $0$ and a standard deviation of $0.1$. Those angles are then converted to 6D in-going and out-going directions as inputs to the networks.

## B.2    Ablation Studies

In Tab. 4 and Fig. 7, we report the ablation results of fitting the BRDF with constraints, including the dot-product kernel using SIREN as the encoder, the hypernet basis with SIREN as the main network, and the constrained layer integrated into SIREN. The hypernet basis uses a SIREN that has a width of 442 as the main network, and a ReLU MLP with one hidden layer with 100 nodes as the hypernetwork. The constraint layer has a width of 319. The hypernet basis and constraint layer produce extremely poor results and are completely unable to perform this task. While the dot-product kernel performs better than both the hypernet basis and the constraint layer, it still significantly underperforms when compared to the Gaussian kernel.

## B.3    Learning Curves

In Fig. 8 and Tab. 5, we report the learning curves, training times, and inference times for various approaches in the BRDF reconstruction experiment. CNF with a Gaussian kernel basis outperforms other approaches in terms of convergence, while maintaining similar training and inference efficiency.

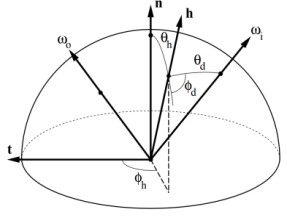

Figure 6: Angles for sampling constraint points for material learning. Figure from [33].

Table 4: Ablations on material appearance learning task. All methods here are based on SIREN [34].

|  | RMSE$_{\times 10^{-2}}$ | PSNR | SSIM |
|---|---|---|---|
| Hypernet Basis | $39.02 \pm 16.92$ | $9.74 \pm 6.41$ | $0.70 \pm 0.06$ |
| Dot-product Kernel | $9.49 \pm 11.32$ | $28.33 \pm 13.22$ | $0.86 \pm 0.13$ |
| Constraint Layer | $41.85 \pm 14.14$ | $8.37 \pm 4.37$ | $0.70 \pm 0.06$ |
| Gaussian Kernel | $\mathbf{0.60 \pm 0.54}$ | $\mathbf{47.95 \pm 7.99}$ | $\mathbf{0.99 \pm 0.01}$ |

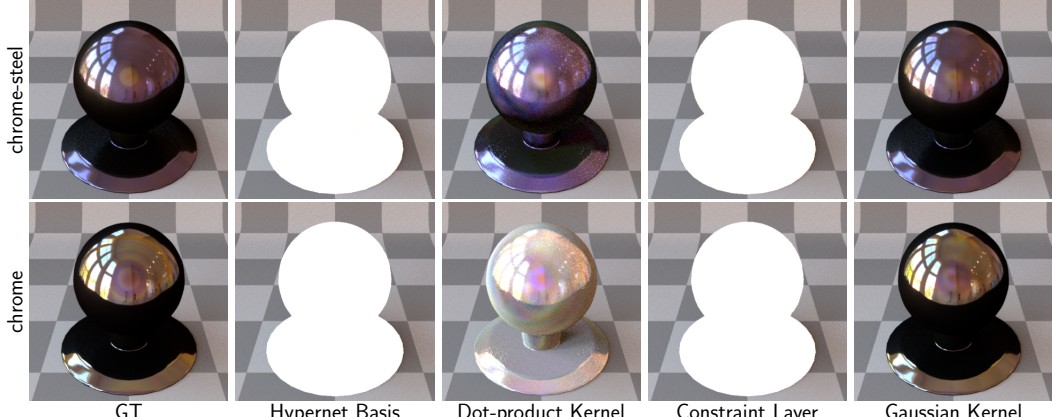

Figure 7: Ablation on the material appearance fitting.

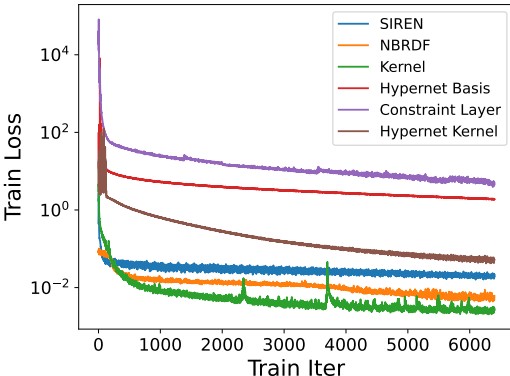

Figure 8: Learning curves of various approaches for BRDF reconstruction.

Table 5: Training and inference time (in seconds) of various approaches for BRBF reconstruction.

| Method | Training | Inference |
|---|---|---|
| SIREN [34] | 578.89 | 0.3348 |
| NBRDF [35] | 19.702 | 0.2966 |
| Constraint layer [26] | 577.49 | 0.2539 |
| Gaussian kernel | 1101.3 | 0.7284 |
| Hypernet basis | 2997.5 | 0.6933 |
| Hypernet kernel | 1784.1 | 1.1692 |

### B.4 Full Results

The complete rendering results for all materials included in the MERL dataset [21] are shown in Fig. 9, Fig. 10, Fig. 11, Fig. 12, Fig. 13, and Fig. 14, and are available on the project page `https://zfc946.github.io/CNF.github.io/`. We also show the SSIM errors along with the renderings.

# C  Interpolatory Shape Reconstruction

## C.1  Experiment Details

### C.1.1  Exact Normal Interpolation

In the experiments on exact normal interpolation, the shapes are represented by the level sets of our hypernet kernel. The encoder of our hypernet is constructed of an MLP, which has a single hidden layers containing 1200 hidden units. Between each layer we use a softplus activation, with $\beta = 10$ and the encoder maps to a feature space of size 800. We use a Gaussian kernel which is initialized with a standard deviation of 50. For all experiments shown, we use a maximum of 50 epochs, a learning rate of $10^{-5}$, and the Adam optimizer. To extract the zero-order level set of the implicit representation we use the *Marching Cubes* algorithm, on a uniform grid sampled at a step size of 1/60. For all shapes except the circle, we pretrain the weights of the hypernet encoder to first represent a circle with uniformly sampled constraints equal to the number of constraints of the target shape. The Eikonal loss function defined in Eq. 13 is defined on the domain $\Omega = [-2, 2] \times [-2, 2]$, and at each iteration we use 1000 points randomly sampled from $\Omega$.

### C.1.2  Large Scale Constraints

In the demonstration of our *patched-based* approach, we reconstruct point clouds with 10,000 uniformly sampled points. Our current implementation of the patch-based solver doesn't support the hypernet kernel, so we regularize the reconstruction problem using a finite difference scheme [39, 42] and impose the point constraints of $\Phi(\boldsymbol{x}_i + \varepsilon \boldsymbol{n}_i) = \varepsilon$ and $\Phi(\boldsymbol{x}_i - \varepsilon \boldsymbol{n}_i) = -\varepsilon$, where we set $\varepsilon = 0.01$ for all reconstructions. As well as the tunable support size, our implementation also places a hard limit on the total number of constraints that are considered for each query point; for all experiments we limit the nearest neighborhood size to 60. Furthermore, as our current implementation does not support tensor batching, the run-time for the patch-based approach, with a marching cubes grid resolution of $[160 \times 160 \times 160]$, is approximately 8 hours.

In the context of sparse solvers, we acknowledge that our current implementation is not completely vectorized for fast GPU execution with batched training points, although the operation of the solver itself is highly efficient (execution within seconds). This limitation primarily arises due to the limited support of sparse matrix operations in popular ML frameworks. However, it is crucial to emphasize that this constraint does not reflect a shortcoming in CNF's design. We hope to attract more attention from the ML community to enhance support for sparse matrix operations within the frameworks to resolve this implementation constraint.

## C.2  Evaluation of Normal Interpolation

To evaluate the merit of our normal interpolation approach for shape representation, we compare shape representation using our exact gradient constraints and pseudo-normal constraints as used in NKF [41] and SIREN [34]. We use the same hypernet kernel approach for all experiments, with the same training setup as Sec. C.1.1, except that we do not pretrain the hypernet encoder to represent a circle. For the pseudo-normal constraints, we set the value of $\epsilon = 0.1$.

In Fig. 15 we show a visual comparison between the reconstructions achieved when using pseudo-normal constraints and exact gradient constraints. To quantitatively assess each approach we also report the mean normal error:

$$\mathcal{E}_{\boldsymbol{n}} = \frac{1}{N} \sum_{i=1}^{N} \|\boldsymbol{n}(\boldsymbol{x}_i) - \hat{\boldsymbol{n}}(\boldsymbol{x}_i)\|, \tag{21}$$

where $\boldsymbol{n}(\boldsymbol{x}_i) = \nabla_{\boldsymbol{x}} \Psi(\boldsymbol{x}_i)$ and $\hat{\boldsymbol{n}}(\boldsymbol{x}_i)$ is the ground truth normal value at $\boldsymbol{x}_i$. The results are tabulated in Table 6.

As is evident both from our qualitative visualization and quantitative evaluation, CNF is able to impose exact normal constraints on neural implicit surfaces during initialization and throughout training (to within floating point accuracy).

|          | SIREN ($*$)          | Pseudo-normal ($\dagger$) | Pseudo-normal ($*$)     | Exact normal ($\dagger$) | Exact normal ($*$)     |
|----------|----------------------|---------------------------|-------------------------|--------------------------|------------------------|
| Circle   | $1.71 \times 10^{-2}$ | 0.776                     | 0.499                   | $4.516 \times 10^{-6}$   | $1.083 \times 10^{-6}$ |
| Line     | $1.02 \times 10^{-3}$ | 3.817                     | $1.589 \times 10^{-2}$  | $2.737 \times 10^{-6}$   | $6.963 \times 10^{-6}$ |
| Triangle | $3.21 \times 10^{-1}$ | 0.466                     | 0.238                   | $7.371 \times 10^{-5}$   | $6.292 \times 10^{-6}$ |
| Diamond  | $1.05 \times 10^{-1}$ | 1.588                     | 0.277                   | $1.450 \times 10^{-5}$   | $3.368 \times 10^{-6}$ |

Table 6: Evaluation of error in the normal vector of the implicit field, as defined in Eq. 21. The symbols $\dagger$ and $*$ refer to measurements taken on initialization and after training the field, respectively.

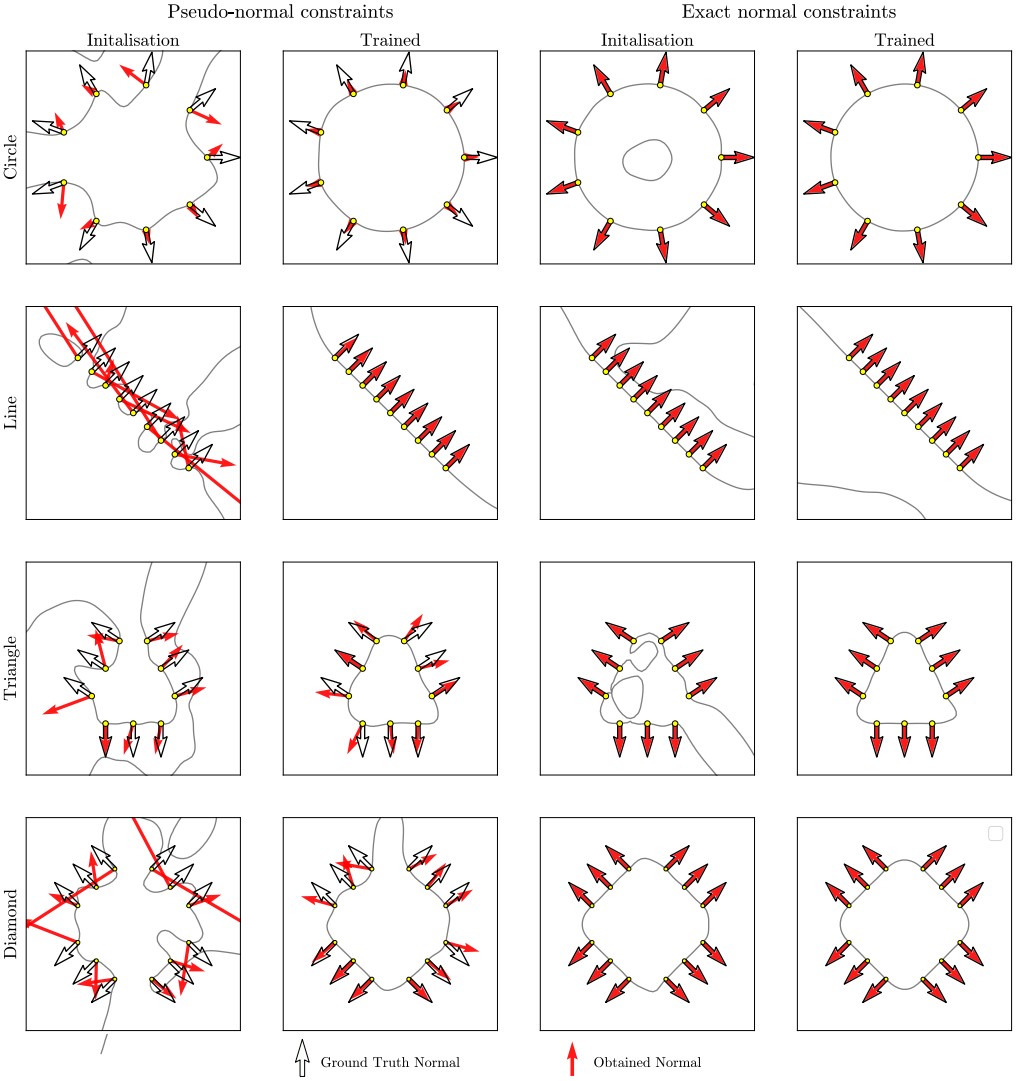

Figure 15: We assess the effectiveness of our precise normal interpolation technique for shape representation. We present the normal vector of the encoded implicit surface (red arrow) before and after training to minimize an Eikonal constraint, for fields bound by pseudo-normal constraints and our exact constraint method. We compare with the ground-truth shape normal vectors (black arrow); our approach produces a surface that faithfully represents both the constraint points and the constraint normal vector.

## C.3 Learning Curves

In Fig. 16 and Tab. 7, we report the learning curves, training times, and inference times for various approaches in the experiment on interpolatory surface reconstruction with exact normal constraints. CNF outperforms prior work in terms of convergence, while maintaining similar training and inference efficiency.

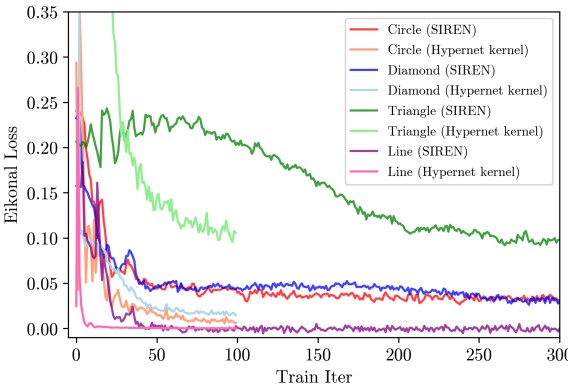

Table 7: Average training and inference time (in seconds) for surface reconstruction with exact normal interpolation.

| Method | Training | Inference |
|---|---|---|
| Hypernet kernel | 242 | 0.0663 |
| SIREN [34] | 148.4 | 0.1437 |

Figure 16: Learning curves of various approaches for surface reconstruction with exact normal interpolation.

# D  Self-tuning PDE Solver

## D.1  Experiment Details

We selected 1D advection (described in Eq. 15 in Sec. 4.4) as a demonstration of the effectiveness of CNF in solving partial differential equations (PDEs). This choice was made due to its simple, linear form. Additionally, one of the advantages is that the ground truth function has a convenient analytical expression, eliminating the need for iterative solvers for validation. Specifically, our objective is to recover the following function:

$$u(x,t) = \sin 2\pi(x - \beta t), \quad x \in (0,1), \ t \in (0,1). \tag{22}$$

## D.2  Learning Curves

In Fig. 17 and Tab. 8, we report the learning curves, training times, and inference times for various approaches in the PDE experiment. We include SIREN [34] as an additional baseline for evaluation. CNF outperforms SIREN in terms of convergence, while maintaining similar training and inference efficiency.

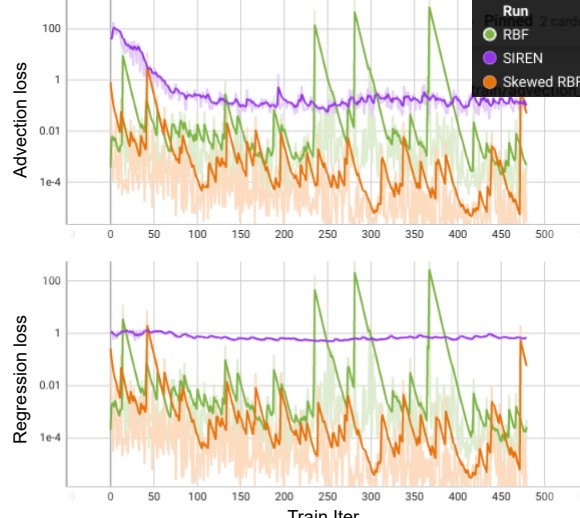

Table 8: Training and inference time (in seconds) of various approaches for self-tuning PDE solver.

| Method | Training | Inference |
|---|---|---|
| SIREN [34] | 36.7 | 0.0087 |
| RBF | 160.7 | 0.0056 |
| Skewed RBF | 265.6 | 0.0061 |

Figure 17: Learning curves of various approaches for self-tuning PDE solver.

## D.3 Transfer Learning

In Sec. 4.4, we detail how CNF can automatically fine-tune the hyperparameters of the skewed RBF on an irregular grid. Here, we illustrate that, given optimized hyperparameters of the skewed RBF for a particular grid, we can employ the resulting skewed RBF to directly solve a different advection equation (such as a different initial condition) at inference time, without additional training, if the grid remains the same.

For demonstration, we incrementally add a shift, $\mu$, to the initial condition of the advection equation, starting from small to large shifts:

$$u_0(x) = \sin(2\pi x) + \mu, \quad x \in (0, 1).\tag{23}$$

The hyperparameters of the skewed RBFs were trained solely when $\mu = 0$. When the initial condition, $u_0(x)$, is shifted, the new advection equation can be solved with the same kernel at inference time. No further training was conducted; see Tab. 9 for the results.

The efficacy of this approach can be explained by our perspective of formulating the constrained optimization problem as a collocation problem with learnable inductive bias. Once the preferred inductive bias away from the collocation points is obtained, solving a PDE at the same collocation points is reduced to a pure collocation problem, which requires minimal to no further training.

Table 9: Demonstration of transfer learning with the skewed RBF kernels. The kernels were initially trained exclusively for a 1D advection function with no shift (first row). Subsequently, their transfer learning capabilities were tested at varying shifts. No training was conducted other than for the first row.

| Shift $\mu$ | Random init Skewed RBF | | Pre-trained Skewed RBF | |
|---|---|---|---|---|
| | RMSE | nRMSE | RMSE | nRMSE |
| 0 | 0.0127 | 0.0227 | **0.0070** | **0.0129** |
| 1 | 0.0460 | 0.0736 | **0.0049** | **0.0074** |
| 10 | 0.5788 | 0.0594 | **0.0177** | **0.0018** |
| 100 | 5.8613 | 0.0587 | **0.2221** | **0.0022** |
| 1000 | 60.143 | 0.0602 | **2.3113** | **0.0023** |
| 10000 | 635.78 | 0.0636 | **23.844** | **0.0024** |

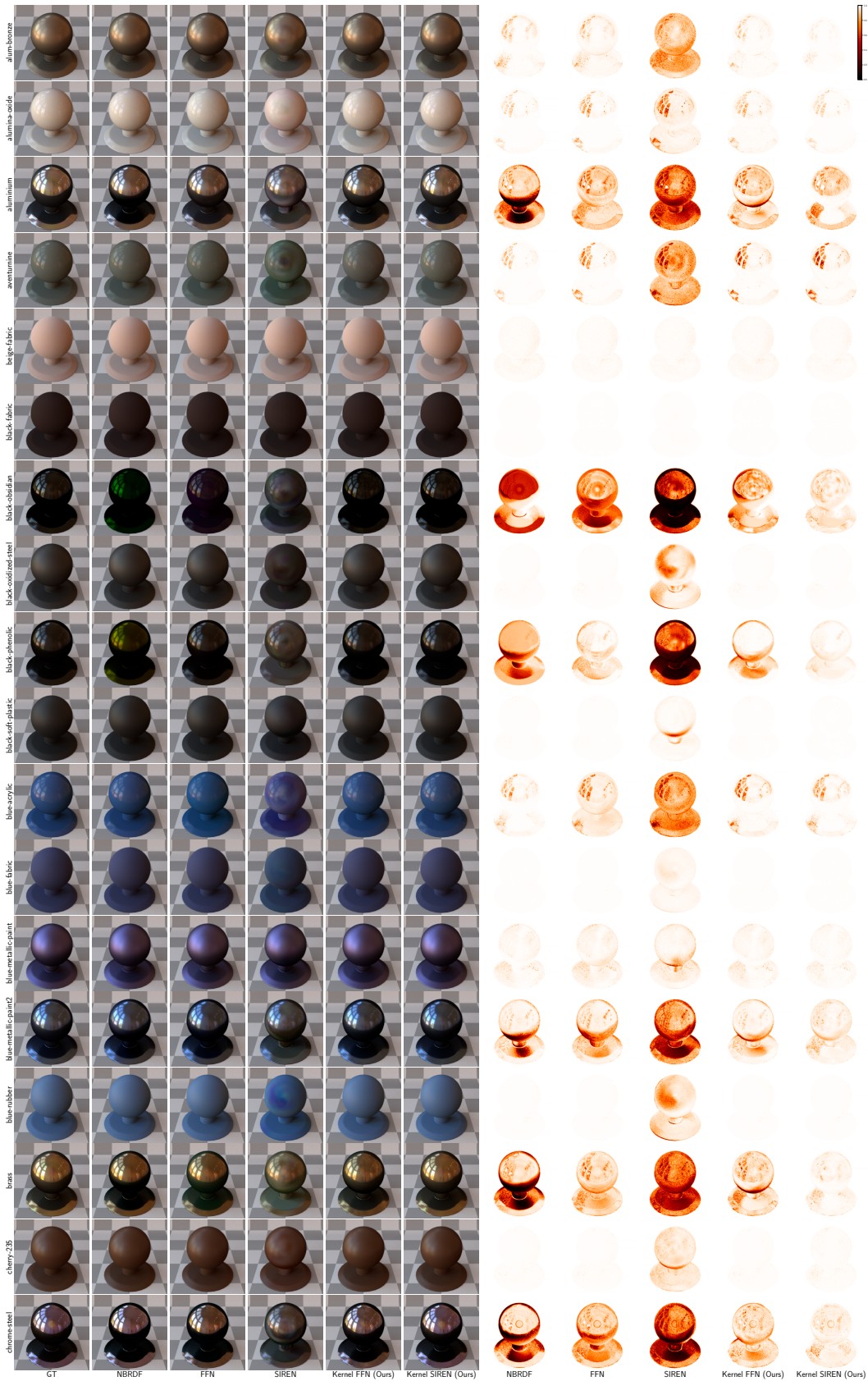

Figure 9: Learned material appearance.

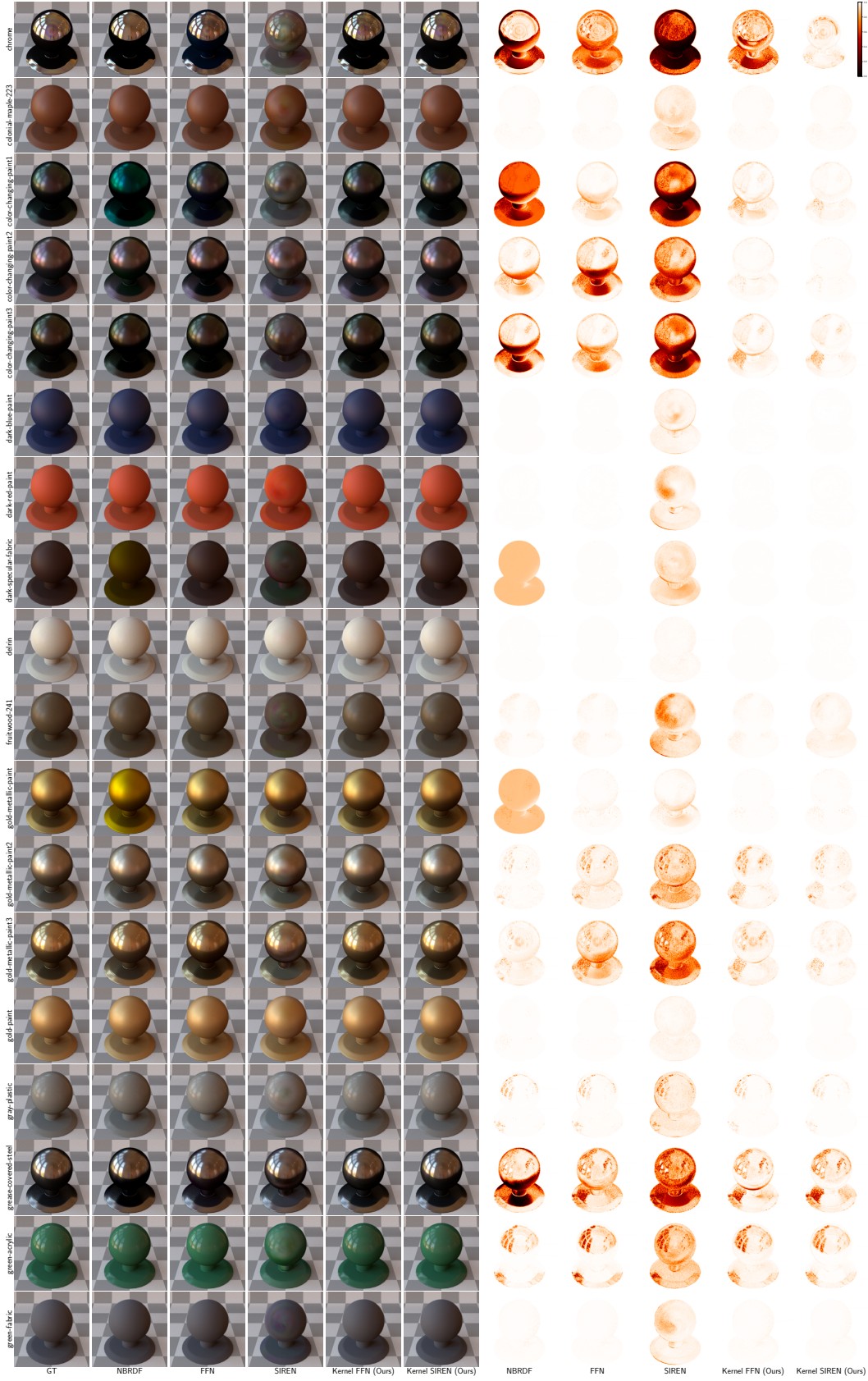

Figure 10: Learned material appearance.

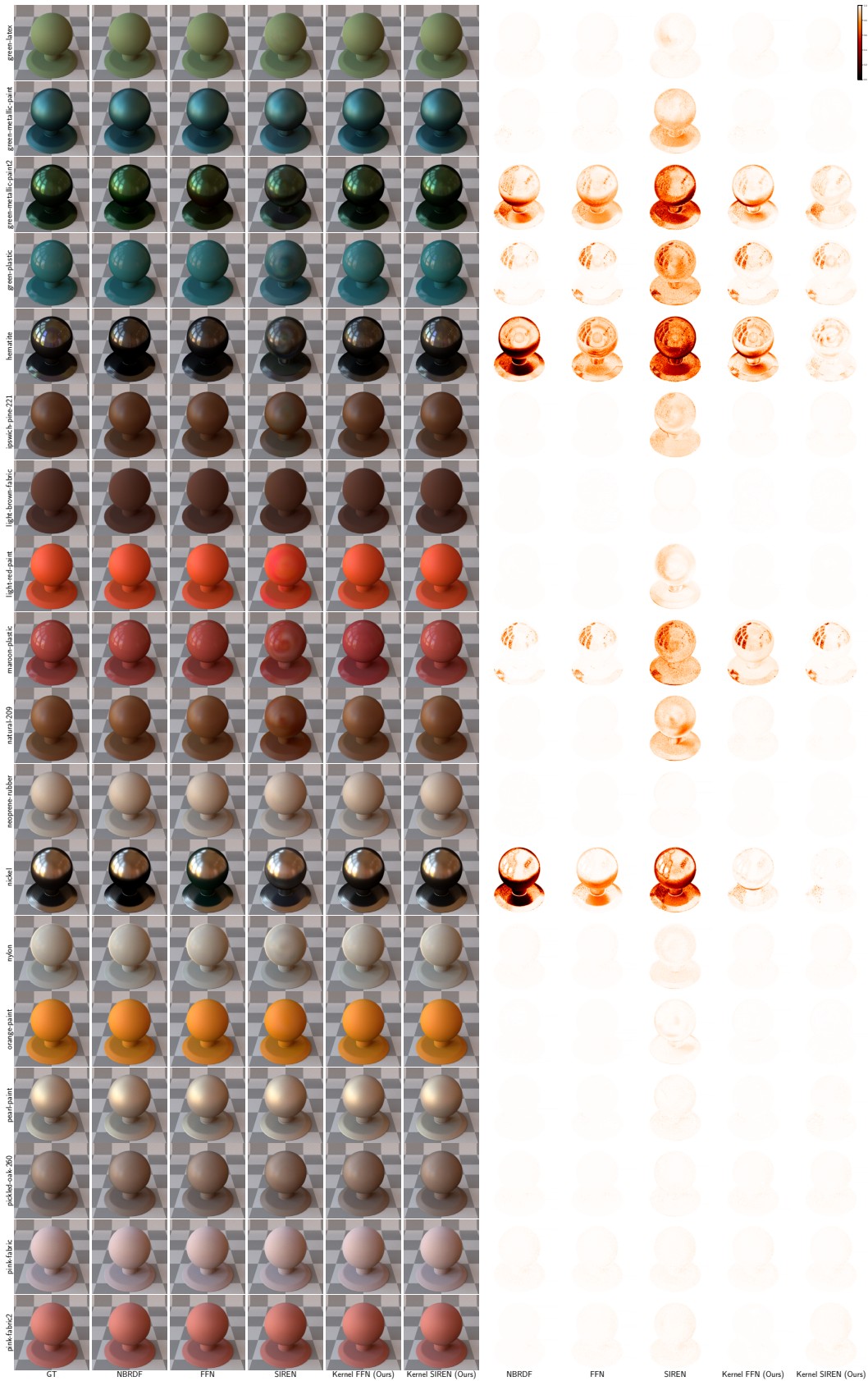

Figure 11: Learned material appearance.

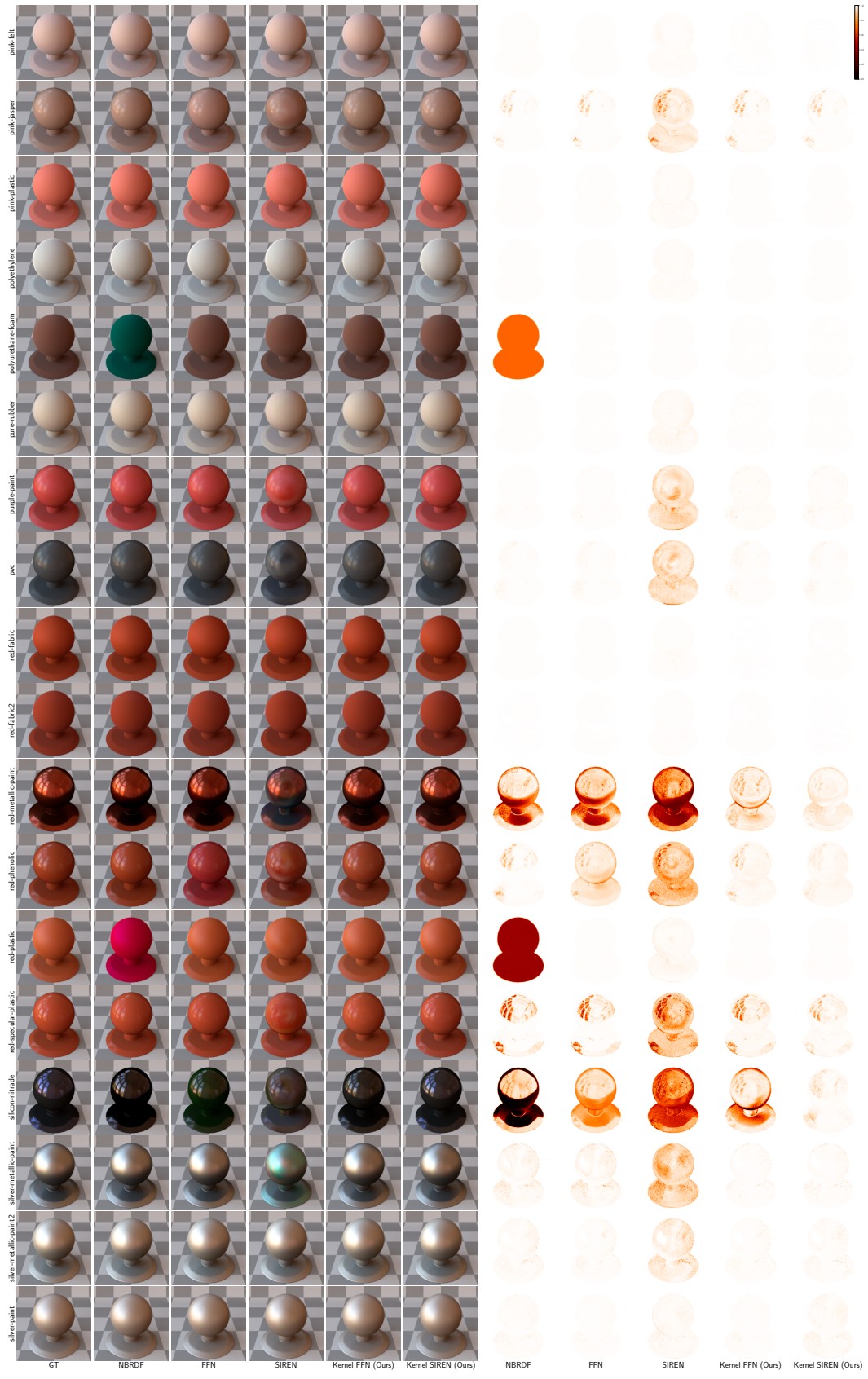

Figure 12: Learned material appearance.

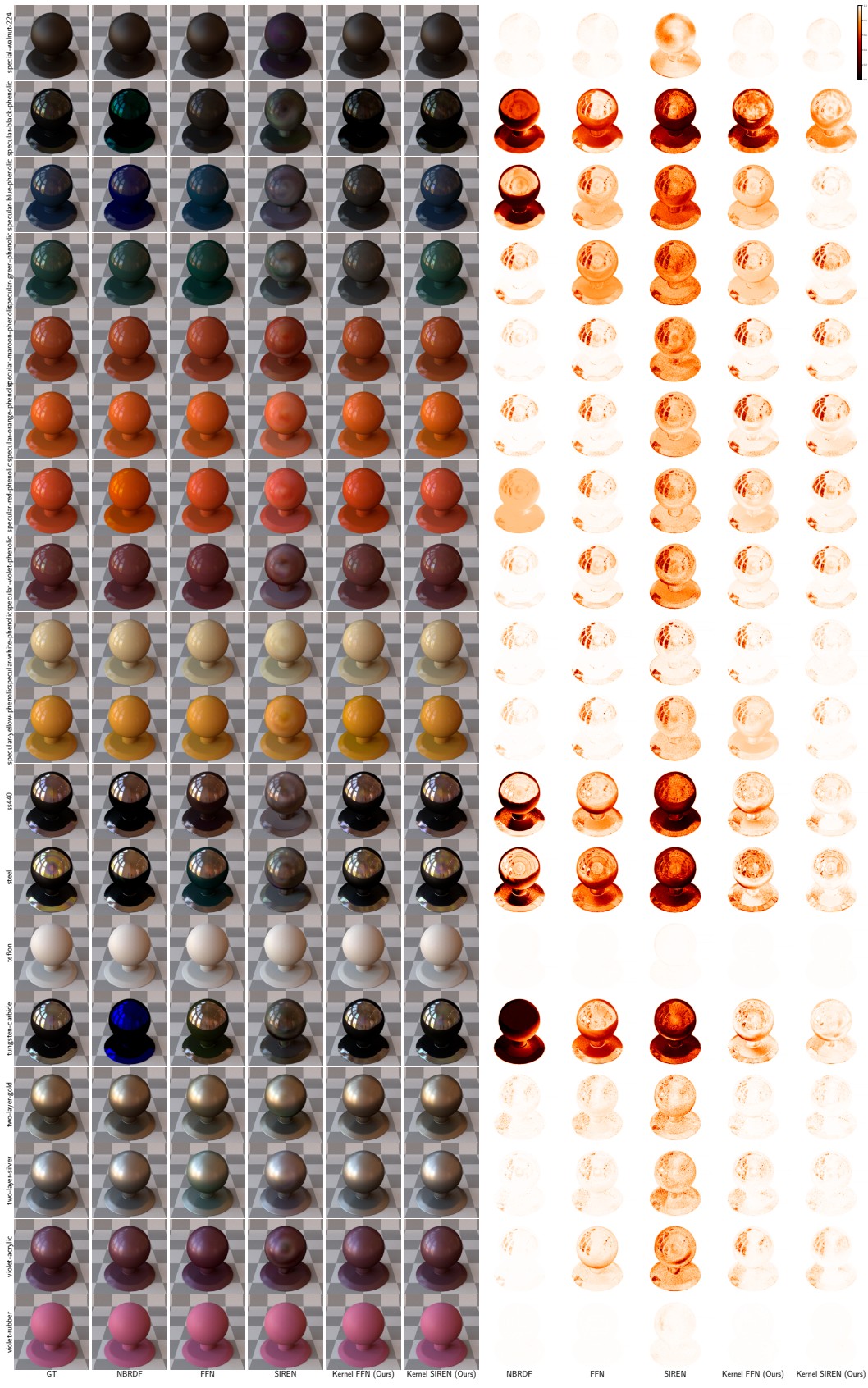

Figure 13: Learned material appearance.

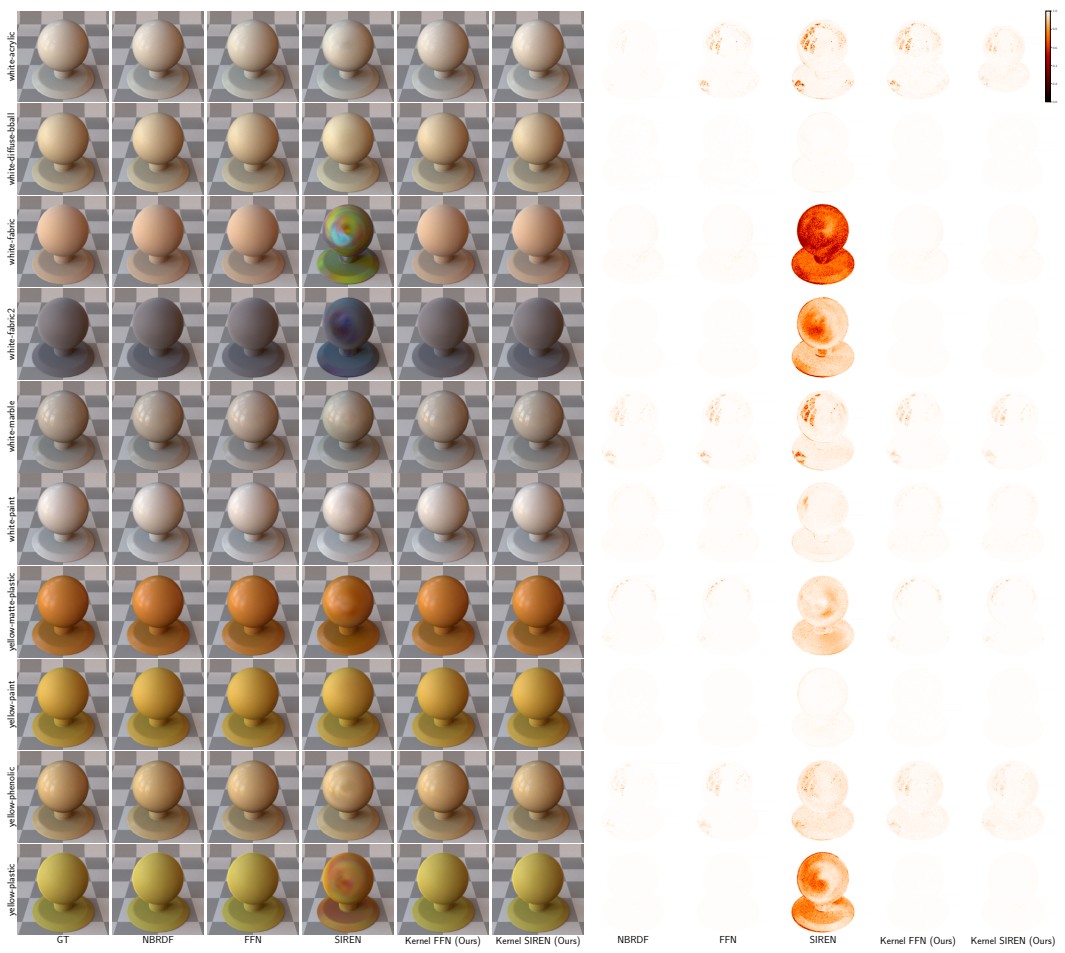

Figure 14: Learned material appearance.

