# OpenReview forum: "Neural Fields with Hard Constraints of Arbitrary Differential Order"
_NeurIPS.cc/2023/Conference — NeurIPS 2023 poster_

### Official Review · Reviewer_o12o · 2023-07-04

**Soundness:** 4 excellent
**Presentation:** 3 good
**Contribution:** 3 good
**Rating:** 6
**Confidence:** 4

**Summary:**

The authors propose a method to enforce hard constraint points on neural fields. Instead of a single black-box coordinate network predicting the field values, this work uses neural networks to learn basis functions which are then combined in a linear transformation. Given enough basis functions, the weights of this linear transformation can be found by solving a system of linear equations.

**Strengths:**

The paper is well-written and easy to understand. The authors have included their source code which seems reasonably well organized.

The method itself is simple and guarantees that constraint points are not violated. It outperforms existing methods on the MERL BRDF dataset.

The paper includes a varied selection of experiments to validate the method, and some ablation studies have been performed and reported in the appendix.


**Weaknesses:**

The paper contributes little in terms of theory. While the derivation is simple enough, it would have been nice to see a theoretical argument for why this method converges faster than unconstrained training, at least for special cases.

The constraints seem to be applicable only to single points, which is not clear from the abstract. Consequently, initial states and boundary conditions cannot properly be handled using this method.

Experiments 4.1 and 4.4 are synthetic and very simple. Only experiments 4.2 and 4.3 measure real-world performance.

The paper introduces six basis functions in chapter 3.2 but they are never compared against each other in the experiments. The authors simply employ different basis functions for different experiments. This makes it hard to judge which one to use for any given problem. Comparisons against state-of-the-art is also sparse. Only the BRDF fitting experiment compares to related work.

The paper does not include learning curves for the various experiments. Figure 5 is somewhat related to learning curves, but the quantitative evaluation of the training process is severely lacking. Training time is not discussed in the paper either. Learning curves against wall-clock time would be appreciated.

Minor:
* L11: The claim in “Our approaches are demonstrated in a wide range of real-world applications.” seems a bit of a stretch. It’s more like one to two.
* L41: Missing citation for the statement about inequality constraints.
* L72-78: The review of previous work is missing stream functions for divergence-free fields (e.g. Deep fluids, 2019) and other conservation properties (e.g. Guaranteed Conservation of Momentum for Learning Particle-based Fluid Dynamics, 2022). Hamiltonian and Lagrangian networks have also been used with conserved properties in mind.
* The citations in the main text do not have hyperlinks to the references at the end.
* The hybrid kernel basis is poorly explained in 3.2.
* Footnote 2 is not explained. It is unclear what kind of LaTeX expressions can be specified.
* Chapter 5 is more outlook than summary
* A.2 and Figure 5: Which experiment does this belong to?


**Questions:**

In Eq. 11, why do you minimize the log L1 loss, not the log L2 loss?

**Limitations:**

While some advantages and disadvantages of various basis functions are mentioned, the authors do not provide a limitations section and only lightly touch on general limitations of the method.

---

> ### Author Rebuttal · Authors · 2023-08-10
>
> Thank you for the constructive feedback.
>
> ## Theoretical analysis
>
> We greatly appreciate this suggestion. Please refer to the general response Q3.
>
> ## Summary of basis functions
>
> Here, we summarize the properties of various basis functions.
>
> |                                                                  | Independent basis | Constraint basis | Hypernetwork basis | Dot-product Kernel basis | Gassian Kernel basis | Hypernetwork kernel basis |  Hybrid kernel basis  |
> |:----------------------------------------------------------------:|:-----------------:|:----------------:|:------------------:|:---------------------------------:|:-----------------------------:|:-------------------------:|:---------------------:|
> |                         Learning capacity                        |                   |       Poor       |        Poor        |                Fair               |              Good             |            Good           |          Good         |
> |                        Linear independence                       |                   |       Poor       |        Fair        |                Poor               |              Good             |   Good (if using Gaussian)  |          Good         |
> |                          Matrix sparsity                         |       Dense       |       Dense      |        Dense       |               Dense               |             Sparse            |  Sparse (if using Gaussian) |         Sparse        |
> |  Number of model parameters independent of number of constraints |         No        |        Yes       |         Yes        |                Yes                |              Yes              |            Yes            |          Yes          |
> |                       Controllable sparsity                      |         No        |        No        |         No         |                 No                |               No              |             No            |          Yes          |
> |                     Higher-order constraints                     |        Yes        |        Yes       |         Yes        |                 No                |               No              |            Yes            | Yes (if using hypernet) |
>
>
> For problems lacking high-order constraints, like the BRDF experiment, we recommend the Gaussian kernel basis. For problems involving high-order constraints, such as sparse shape reconstruction (line 239-251), we suggest using the hypernetwork kernel basis. For large-scale tasks, such as dense shape reconstruction (line 252-255), we recommend the hybrid kernel.
>
> Please refer to Sec. 3.2 for relevant explanations, and supplementary A2 and B2 for empirical comparison.
>
> ## Single point constraints
>
> We agree that our method applies to discrete points rather than a continuous set. However, CNF is perfectly suitable for initial and boundary value problems, as demonstrated in Sec 4.4. In fact, all the major PDE solvers such as FDM, FEM, and spectral methods solve the initial and boundary problems at discrete grid points. CNF can improve the performance of the Kansa method, a type of spectral method for solving general PDEs.
>
> Compared to mesh-based solvers such as FDM and FEM, the Kansa method has the advantage of solving a PDE on an irregular grid without meshing, i.e. grouping the points into triangles or quadrilaterals. CNF addresses a major limitation of the Kansa method (please refer to line 260-266). We plan to perform a more extensive evaluation of CNF in comparison to other PDE solvers in future work.
>
> ## Hybrid kernel basis
>
> The hybrid kernel was designed to promote the sparsity of the matrix in Eq. 4.
>
> $\Psi_i\left(\mathbf{x}\right) = \kappa\left(\phi_i\left(\mathbf{x}_i\right), \phi_i\left(\mathbf{x}\right)\right) \kappa_G\left(\mathbf{x}_i ,\mathbf{x}\right)$
>
> The first part $ \kappa\left(\phi_i\left(\mathbf{x}_i\right), \phi_i\left(\mathbf{x}\right)\right)$ could be either the Gaussian kernel basis or the hypernet kernel as defined in Sec. 3.2 (depending on the task). We multiply it with a compactly supported function $\kappa_G$ so that the matrix sparsity can be explicitly adjusted. A candidate $\kappa_G$ can be a truncated Gaussian kernel such as:
>
> $
>     \kappa_G\left(\mathbf{x}_i, \mathbf{x}\right)  =  \begin{cases}
>        \exp\left( -\frac{\|\mathbf{x}_i- \mathbf{x}\|^2}{2\sigma^2}\right) & \text{if } \|\mathbf{x}_i- \mathbf{x}\| < 3\sigma\\
>         \text{ or } 0 & \text{if } \|\mathbf{x}_i- \mathbf{x}\| \ge 3\sigma.
>     \end{cases}
> $
>
> Reducing $\sigma$ yields a matrix with more zero entries.
>
> ## Comparisons against SOTA
>
> Besides the BRDF fitting experiment, our shape reconstruction experiment also compares with the state-of-the-art work, NKF. It is important to note that, unlike our work, the design NKF cannot model normals with hard constraints. Please refer to supplementary A1 for theoretical explanation and C2 for empirical evaluation.
>
> ## Footnote 2
>
> This highlights our user-friendly interface for defining complex, higher-order linear operators (source code in `diff_utils.py` in supplementary):
> ```
> def compute_op(latex_str, y, x)
> ```
> In the context of the advection operation $\frac{\partial f(x, t)}{\partial t} + \beta(x) \frac{\partial f(x, t)}{\partial x}$, our interface can parse `latex_str` if provided as `f_{x_1} + {beta}f_{x_0}` and compute this operation automatically. The usage of this interface will be further clarified through illustrative examples, which will be provided alongside the code release.
>
> ## A.2 and Figure 5
>
> This is a standalone experiment where we minimize the condition number of the matrix $A_f$ in Eq. 4 given various basis functions. This experiment measures the inherent linear independence of basis functions of different designs.
>
> ## log L1 loss
>
> We follow the prior work NBRDF, which also uses log L1, for a fair comparison. BRDF values have a large variation in scale (0.1-700). To correctly fit both ends, logL1 provides a good balance as reported by the NBRDF paper.

---

> > ### Comment · Reviewer_o12o · 2023-08-13
> >
> > Thank you for clarifying! I believe the learning curves and table comparing the different basis functions are a valuable addition.
> >
> > **Single point constraints**
> >
> > While it is true that classical methods also use boundary conditions at discrete points, these methods have a clearly defined and interpretable interpolation scheme. Fields defined through coordinate networks effectively use black-box interpolation between the constraint points. Can you set an upper bound on the deviation from the given boundary condition for all boundary points (constraint points and interpolated points)?

---

> > > ### Author Response · Authors · 2023-08-15
> > >
> > > Thank you for the suggestions. We will incorporate the additional reports into the revised paper.
> > >
> > > While classical methods such as FDM may employ interpretable interpolation schemes for continuous set evaluation, this interpolation requires a well-defined meshing of grid points – grouping the points into triangles or quadrilaterals. Note that, for these methods, meshing is required for both solving the PDE at the grid points and evaluating the solution away from the grid points. Meshing on an irregular grid, in the case of our experiment in Section 4.4, proves to be extremely challenging and severely harms the accuracy of these methods. In contrast, our approach, as is based on Kansa, allows for analytical and continuous evaluation of the solution function across the entire domain, without requiring any interpolation or meshing.
> > >
> > > Our PDE solver also did not use coordinate networks but a proposed skewed RBF as the basis function. We only suggest using neural networks when the behavior of the basis functions to be optimized away from the constraint points is highly complex, as is in the BRDF and shape-reconstruction experiments. When it comes to solving PDEs, the only prior we need is linear independence and smoothness. Therefore, our proposed skewed RBF, which is essentially another variant of Gaussian kernel basis but without the neural encoder, is sufficient. We will also highlight this in the summary of basis functions.
> > >
> > > The objective of our PDE experiment was to demonstrate that we can address a major limitation of Kansa — tuning the hyperparameters of the basis function.
> > >
> > > There have been studies [1] conducted to establish the error bounds of Kansa. The error estimate of Kansa is largely dependent on the selection of the basis functions and their shape parameters. CNF effectively refines the shape parameters to achieve a reduced error range, as evidenced in our empirical comparison. Note that the RMSE reported in Section 4.4 and Supplementary D2 were measured away from the grid points. We leave the rigorous derivation of the exact error bounds of our approach to future work, where we plan to conduct a more comprehensive evaluation of CNF against other PDE solvers.
> > >
> > > [1] Kazemi, B.F., Jafari, H. Error estimate of the MQ-RBF collocation method for fractional differential equations with Caputo–Fabrizio derivative. Math Sci 11, 297–305 (2017).

---

### Official Review · Reviewer_rctK · 2023-07-05

**Soundness:** 3 good
**Presentation:** 4 excellent
**Contribution:** 3 good
**Rating:** 7
**Confidence:** 4

**Summary:**

A broad range of problems can be formulated as linearly constrained problems, e.g., learning material appearance, interpolatory reconstruction, solving linear PDE, etc. In order to solve linearly constrained optimization problems, this paper developed a novel hard constraint method that builds upon neural fields and differentiable linear solver, naming constrained neural fields (CNF).

**Contribution**:
1. A novel methodology CNF is proposed, specifically, linear equality constraints are transformed into a linear system, i.e., eq. (3)-(4);
2. Then both weights of neural fields, $\beta_i$, and learnable parameters, $\theta$, of neural fields (see eq. (2)) can be learned from gradient descent of the objective function in eq. (1) given a differentiable linear solver is applied to eq. (3)-(4);
3. Hyper kernel basis is proposed, which is benchmarked with various basis functions (see section 3.2) and demonstrates advantages such as stable conditional number throughout training (Fig. 5);
4. In experiments, 4 examples from very different background are solved by CNF with superior performance, showing the potential of CNF as a general learning framework for linearly constrained problem.

**Strengths:**

**Originality**: CNF is a novel method of solving linearly constrained problem by implementing neural fields based on differentiable linear solver.

**Quality**: The paper shows that CNF can solve various problems with high performance.

**Clarity**: The methodology is clearly presented. Worthy of mentioning, the analysis of conditioning of the matrix due to different kernel methods (see Appendix A) is quite convincing about why the authors believe Gaussian kernel with hybrid kernel basis (see eq. (9)) is the best choice.

**Significance**: This work can be applied to a broad range of linearly constrained problems.

**Weaknesses:**

The work relies on differentiable linear solvers, and therefore the most suitable problem for CNF is a linearly constrained problem. In section 5, the authors discussed that CNF can be applied to nonlinear problems given a differentiable solver. However, such a differentiable solver to nonlinear problems is in general not easy to get. Hence, CNF is currently limited to linear problems.

**Questions:**

N.A.

**Limitations:**

N.A.

---

> ### Author Rebuttal · Authors · 2023-08-10
>
> We greatly appreciate the positive comments and feedback.
>
> We acknowledge the challenges posed by nonlinear problems, particularly in terms of convergence and the expensive computational graph of nonlinear solvers. Potentially, the latter could be addressed through the use of implicit layers. We plan to further study it in future work.

---

### Official Review · Reviewer_RAef · 2023-07-05

**Soundness:** 3 good
**Presentation:** 2 fair
**Contribution:** 3 good
**Rating:** 6
**Confidence:** 3

**Summary:**

The paper presents a method for integrating hard constraints, represented by a linear operator, into neural field basis functions. This is achieved by learning kernel functions as basis functions at specific constraint points. Through experimentation, the paper provides evidence to show the effectiveness of the proposed method in comparison to unconstrained neural fields across diverse practical tasks.

**Strengths:**

Originality: The paper introduces a novel approach by directly applying linear operator constraints to the basis functions constructed with neural field functions and learning a linear representation. Also, the weights are found by applying a solver to a linear system, which is nice and removes some optimization problems when the weights should also be learned.
Quality: The results obtained in the paper demonstrate the effectiveness of the proposed method across various tasks. I appreciate the efforts in comparing different common implementations of neural field bases.
Clarity: The paper effectively explains the reasoning behind critical implementation choices, such as the selection of basis functions and the choice of kernel.
Significance: The proposed approach addresses the significant challenge of the application of explicit hard constraints for neural fields.

**Weaknesses:**

- As a reader, I found it challenging to comprehend the training procedure without delving into the code. Therefore, providing a comprehensive explanation would facilitate understanding. This would make it easier to grasp the methodology.
- Furthermore, sharing more details about how regularization is applied would provide valuable insights into the approach. Explaining the specific methods used for regularization and their impact on the model's performance would enhance the clarity and comprehensiveness of the paper.

**Questions:**

- To enhance clarity, it would be better for the authors to dedicate a separate section to explaining the model training process in detail. This section needs to explicitly highlight which parameters or weights are optimized and specify the optimizer/solver employed for this purpose.

**Limitations:**

The authors have addressed the limitations of the work in the Summary section.

---

> ### Author Rebuttal · Authors · 2023-08-10
>
> Thank you for your valuable feedback.
>
> ## Details regarding the training procedure
>
> We appreciate your suggestion to dedicate a separate section to explain the training process in detail. Please refer to the general response Q1 for training details.
>
> ## Regularization
>
> The only regularization we recommend is a term containing the condition number of the matrix in Eq. 4. The condition number is a standard metric in numerical linear algebra that measures the sensitivity of the solution to errors in the input data, such as matrix coefficients or the right-hand side vector. A singular, noninvertible matrix has an infinite condition number.
> A smaller condition number corresponds to a smaller error in the solution satisfying the hard constraints.
>
> The condition number can be added as a regularization term to the main loss. It can be computed by the ratio of the maximal and minimal singular values of the matrix from SVD decomposition.
>
> Smoothness is also a commonly preferred quality of neural fields, depending on the task. Smoothness can be promoted by adding a total variation term.

---

> > ### Comment · Reviewer_RAef · 2023-08-21
> >
> > Thank you for addressing my concerns. I will maintain my initial rating.

---

### Official Review · Reviewer_t12b · 2023-07-10

**Soundness:** 2 fair
**Presentation:** 3 good
**Contribution:** 2 fair
**Rating:** 4
**Confidence:** 4

**Summary:**

The authors look at enforcing hard constraints on neural fields. Here, the problem formulation is to take continuous coordinates as input and predict the solution on these points as output. The neural field is represented as a linear sum of basis functions, and specifically, variants of a neural kernel function. The constraints must be linear operators which are then satisfied via using a linear solver.

**Strengths:**

- Enforcing constraints more precisely on neural fields could improve prediction performance, and trying to do this via harder constraints seems promising.

- The method utilizes flexible representations of neural fields (i.e., neural kernel fields; as well as other representations of the basis functions) to enforce the relevant constraints.

**Weaknesses:**

- The evaluation metrics are limited, and it is hard to contextualize these results with respect to other neural field methods. For example 1, the authors compare to other neural representation methods, but what about comparing to “soft constraint” approaches as well? It would be helpful to know how speed vs accuracy compares when enforcing the constraint is a softer way.

- There is no discussion of the speed and training time to implement this hard constraint approach. It seems like it would be expensive to do a linear solve on these matrix systems, greatly hindering efficiency. More details about the linear solver would be helpful (as well as how the authors treat the system as fully differentiable).

- One major limitation of this approach is that only linear operators are used as constraints (thereby being able to utilize eon 3). The practicality of this method seems especially useful when the operator is non-linear.

- It would be helpful to describe the problem formulation for each problem (inputs/outputs, form of constraint, what is being minimized, etc.), as this is not always clear from the paper.

**Questions:**

- Is the constraint enforced at both training and test time?

- What linear solvers are being used here? The authors mention “note that when employing general solvers such as SQP and DC3…” but don’t give details about the solvers themselves.

- What is the training time and inference time, and how does it compare to other NN fitting approaches? In particular, these linear solvers could be very expensive.

- Can you show an example of using the method when the constraint is a non-linear operator?

- In example 4.3, is eqn. 12 (Eikonal equation) being solved in a “soft constraint” way? Where are the hard constraints being enforced?

- The format of some of the references needs to be corrected. For example, references 7, 16, 25, 39 do not specify a publication venue.

**Limitations:**

- Limitations are not discussed (besides that this approach does not work for non-linear operators), but it seems like this approach would also be difficult to scale up for larger systems because of the expensive linear solves on larger systems.

---

> ### Author Rebuttal · Authors · 2023-08-10
>
> Thank you for the constructive feedback.
>
> ## Comparison to soft constraint approaches
>
> In our experiment on learning material appearance described in Sec. 4.2, we compared CNF with FFN [35] and SIREN [32], two representative soft constraint methods. Our approach surpasses them qualitatively and quantitatively. Complete results are available in the supplementary (Section B3).
>
> Furthermore, we conducted extra evaluations by comparing CNF with SIREN in shape-reconstruction and PDE-solver experiments. CNF excels in both cases; please refer to Q2 in the general response for details. For a theoretical argument on CNF's superiority over soft approaches, please see Q3 in the general response.
>
> ## Training and inference efficiency, large-scale system
>
> Detailed insights into training duration, inference speed, and learning curves are available in the general response (Q2). CNF is efficient in both training (minutes) and inference (seconds).
>
> The design of CNF also effectively handles large-scale problems. We developed a hybrid kernel to ensure matrix sparsity (line 163-170) and a patch-based sparse solver (introduced in lines 181-185 and evaluated in line 252-255) tailored for this purpose.
>
> Importantly, as CNF weights, $\beta$, can be precomputed, inference does not require solving matrix equations. This further enhances its applicability to large-scale problems. For additional inference procedure details, please see general response (Q1).
>
> In the context of sparse solvers, we acknowledge that our current implementation is not completely vectorized for fast GPU execution with batched training points, although the operation of the solver itself is highly efficient (execution within seconds). This limitation primarily arises due to the limited support of sparse matrix operations in popular ML frameworks. However, it is crucial to emphasize that this constraint does not reflect a shortcoming in CNF's design. We hope to attract more attention from the ML community to enhance support for sparse matrix operations within the frameworks to resolve this implementation constraint. We will include this discussion in the revised paper.
>
> ## Details about the linear solver
>
> For typical cases, we solve the matrix equation through LU decomposition with partial pivoting and row interchanges, which is fully differentiable if the matrix is full rank. Please see general response Q1 for details on how the linear solver is integrated in training.
>
> For large systems, we use the hybrid kernel basis to ensure matrix sparsity (line 163-170) and a patch-based sparse solver (line 181-185) for solving large sparse matrix systems.
>
> ## Practicality of solvers to linear operator constraints vs nonlinear operator constraints
>
> The linear operator constraints cover a wide range of real-world problems, such as interpolating fixed points, fitting exact differentials, and solving linear ODEs and PDEs. Our performance is the best compared to all the prior approaches on the same linear problems.
>
> Most importantly, a general solution to nonlinear problems is typically not the ideal solver for linear problems, which typically require special treatment to improve efficiency. For example, the simplex method as a solver for linear programming typically takes less iterations and memory compared to a general solver. Therefore, linear and nonlinear problems are two separate problems requiring different solutions. A nonlinear solver cannot be a universal solution for both scenarios.
>
> Therefore, we focus on linear problems, and leave nonlinear problems as separate future work.
>
> CNF can be extended to non-linear operator constraints if step 1 of the training process (please refer to the general response Q1) is replaced with a nonlinear solver to solve a nonlinear version of Eq 3. However, such a nonlinear solver would not be an ideal solution for linear operator constraints, as its convergence and efficiency are both difficult to analyse and promote.
>
> ## Problem formulation of each experiment
>
> We are happy to provide detailed math formulation of each experiment in the author-reviewer discussion period. Unfortunately, the rebuttal does not allow for a sufficient word limit to include these details.
>
> ## Is the constraint enforced at both training and test time?
>
> Yes, please refer to the training and inference details in Q1 of the general response for how the constraint is enforced at both training and test time.
>
> ## Discussion of SQP and DC3
>
> Please refer to general response Q3 for details.
>
> ## Show an example of using the method when the constraint is a non-linear operator?
>
> CNF can be extended to non-linear operator constraints if step 1 of the training process (please refer to the general response) is replaced with a differentiable nonlinear solver to solve a nonlinear version of Eq 3. Any iterative method that converges, such as Gauss-Newton, can be a candidate solver. However, the challenge lies in the convergence and the costly computational graph of such iterative solutions. The latter can be potentially addressed through the use of implicit layers [12], which we leave as future work.
>
> ## Is the Eikonal equation being solved in a “soft constraint” way? Where are the hard constraints being enforced?
>
> That is correct, as the Eikonal equation is a non-linear PDE we choose to introduce its geometric bias into the optimization as a soft constraint via training. In this case, the hard constraints are on the points and the point normals themselves - that is, for all $x$ and $n(x)$ in our point set $P$, $F(x) = 0$ and $\nabla F(x) = n(x)$.

---

> > ### Comment · Reviewer_t12b · 2023-08-12
> > **Response to authors**
> >
> > Thank you to the authors for your responses, I appreciate it. I have gone through the responses and the paper again, as well as looked at all the related work.
> >
> > A major limitation of this work is that it is currently only used for linear operators, and that means that it cannot tackle a number of more realistic systems at this point. I know that the authors describe that another solver, like Gauss-Newton, can be used, but it is challenging to converge and the computational graph is expensive. Thus, it seems like this approach is limited because it cannot handle a very large number of important use cases. One of the related constraint works, PDE-CL, seems to be able to work on non-linear differential operators.
> >
> > Edit (2 hours after the above comments):
> >
> > The other point I want to make about training and inference speed is that right now, you are only comparing to other neural network approaches. For a number of these problems, you should also be comparing to classical numerical methods.

---

> > > ### Author Response · Authors · 2023-08-14
> > >
> > > Thank you for your comment.
> > >
> > > ## Nonlinear operator constraints
> > >
> > > We acknowledge that solving problems with nonlinear operator constraints is not the focus of this paper, as previously discussed in the limitations section. However, within the realm of scientific computing, it is crucial to assess the limitations of different computational methods from a holistic standpoint, as elaborated upon in our rebuttal.
> > >
> > > For example, consider the problem of matrix decomposition for solving linear systems. The Cholesky decomposition is highly efficient and stable compared to the more general LU decomposition, but it is limited to positive definite matrices. However, it would be unjust to argue that LU is superior to Cholesky due to its broader applicability, or that Cholesky is superior to LU due to its efficiency and stability. Cholesky and LU were tailored for distinct objectives.
> > >
> > > The objective of our paper was to devise highly efficient and stable methods for addressing linear operator constraint problems. Our approach outperforms existing works in such scenarios, as was substantiated in both theoretical analysis and empirical evaluations.
> > >
> > > There are also several facts to highlight:
> > > - PDE-CL also focuses on linear problems. Their extension to nonlinear problems is no different from what we have suggested (replacing the linear solver with a non-linear least-squares solver) and shares the same challenges as we described. Their evaluation of nonlinear problems is limited to the Burgers equation.
> > > - We have extended the formulation of PDE-CL to a broader context and showed improvements in their design; please refer to the general response Q3.
> > > - As PDE-CL is a concurrent work, a comprehensive empirical comparison with it is infeasible.
> > >
> > > ## Comparison to classical numerical methods
> > >
> > > Please refer to “comparison to general approaches” in the general response Q3 for a theoretical analysis of CNF’s superiority compared to classical methods. This applies to almost all types of classical numerical methods for constrained optimization. We will incorporate this analysis in the revised paper.
> > >
> > > A fair comparison of classical numerical methods would involve solving the constrained optimization problem in the form of the equation in general response Q3. This is computationally infeasible for most classical methods due to 1) the high dimensionality induced by $\theta$ and 2) the nonlinearity due to its formulation. Thus, we follow the standard practice and only present training and inference times w.r.t other neural network baselines.

---

> > > > ### Comment · Reviewer_t12b · 2023-08-14
> > > > **Response to authors**
> > > >
> > > > Thank you for the response.
> > > >
> > > > **Non-linear operator constraints**
> > > >
> > > > While you have suggested replacing the solver for non-linear problem, my point is that it looks like PDE-CL was still able to do a non-linear problem. It sounds like here, your approach is so computationally expensive that you cannot even do a simple non-linear problem.
> > > >
> > > > **Comparison to classical numerical methods**
> > > >
> > > > When you solve, for example, PDEs, the comparison shouldn't be to solving a constrained optimization problem numerically. Obviously, the reason people do not do this for PDEs is because it is expensive, and so we have instead come up with a set of numerical solver tools to solve these systems more quickly instead. Thus, the comparison should be to use one of these standard numerical solvers. For the examples you have shown in the paper, a numerical solver would be very quick, and it sounds like it may be faster than your method.

---

> > > > > ### Author Response · Authors · 2023-08-15
> > > > >
> > > > > Thank you for the clarification.
> > > > >
> > > > > The computational complexity of ours and PDE-CL are the same when it comes to differentially solving nonlinear equations. Most importantly, our version of the PDE solver is, in essence, a Kansa method, which is a classical collocation method that has been proven to be effective for solving general PDEs including **nonlinear** ones.
> > > > >
> > > > > The objective of our PDE experiment was to demonstrate that we can address a major limitation of the classical Kansa method, tuning the hyperparameters. We formulate hyperparameter tuning as a constrained optimization problem. Once the optimal hyperparameters are obtained, the PDE solver reduces a classical collocation method without requiring any training (please refer to supplementary D2 for additional results). The solver itself only takes 0.0061 seconds in the advection case.
> > > > >
> > > > > The results in Sec 4.4 and Supplementary D2 indicate that CNF indeed improves the performance of Kansa. We did not aim to demonstrate that our approach achieves state-of-the-art performance among all the PDE solvers. Therefore, we only compare it with the untuned Kansa method rather than other methods. However, note that our PDE was solved on an irregular grid. This **cannot** be achieved by the majority of the classical methods that are mesh-based, where the grid points need to be grouped into triangles or quadrilaterals, including the most popular FDM and FEM.

---

### Author Rebuttal · Authors · 2023-08-10

We thank the reviewers for the detailed and constructive feedback. Below are our responses to the common questions:

# Q1. Training and inference details

We offer a thoroughly tested codebase that assists users in modeling challenging constraints using CNF. To ensure comprehensiveness, we will also incorporate the following descriptions and pseudocode blocks in the supplementary.

Recall the training objectives

$
\underset{\theta}{\arg\min} $ $\mathcal{L}\left(f_\theta;\theta\right) \quad \text{s.t. }
\mathcal{F} \left[f_\theta\right] \left(\mathbf{x}\right)
= g\left(\mathbf{x}\right) \;\forall \mathbf{x} \in \mathcal{S}:= \left\lbrace x_i\right\rbrace_{i=1}^I,
$

where
$
f_\theta\left(\mathbf{x}\right) = \sum_i \beta_i \odot \Psi_i\left(\mathbf{x}\right),
$

and $\theta$ indicates the learnable parameters of each basis function $\Psi_i$.

The following procedure describes the training process:

```algorithm
Repeat:
    1. Compute 𝜷 by solving Eq. 4
    2. Compute gradient ∂𝐿/∂𝜃 = ∂𝐿/∂𝑓 ∂𝑓/∂𝜃
    3. Update 𝜃 via gradient descent
Until converged
```
Step 1 computes the weights $\mathbf{\beta}$ to ensure that the constraints are always satisfied throughout training. Steps 2 and 3 update the training parameters $\theta$ to optimize the training loss $\mathcal{L}$ under the constraints.

$\mathbf{\beta}$ is computed using an LU decomposition with partial pivoting and row interchanges, which is fully differentiable if the matrix in Eq.4 is full rank.
The computation of $\mathbf{\beta}$ constructs a computational graph that tracks $\frac{\partial \mathbf{\beta}}{\partial \theta}$.

Any loss function with a valid gradient $\frac{\partial \mathcal{L}}{\partial f}$ can be smoothly integrated into our training process.

Next, we have the inference algorithm:

```algorithm
Input: 𝑥
Output: 𝑓_𝜃(𝑥)

If 𝜷 is None:
    Compute 𝜷 from Eq. 4
else:
    𝑓_𝜃(𝑥) ← Σᵢ 𝜷ᵢ ⊙ Ψᵢ(𝑥)
```
Here, $\mathbf{\beta}$ only needs to be pre-computed once since it does not depend on the evaluation point $\mathbf{x}$. As a result, performing inference with CNF is very efficient and boils down to computing a linear combination of $I$ basis functions, a task that can be vectorized for efficiency.

# Q2. Training/inference time, learning curve, and additional evaluation

Please refer to the attached PDF for details. CNF is efficient in training (minutes) and inference (seconds). We also compare with SIREN [32] for shape reconstruction and solving PDEs. CNF demonstrates superior performance in all experiments compared to SIREN.

# Q3. Theoretical analysis

Here, we provide a summary of the theoretical argument that elucidates CNF's superior performance over existing solutions:

### Comparison to general approaches

Many general algorithms in scientific computing, including the popular SQP and more recent DC3 [11], adopt a formulation for the constraint optimization problem as:

$
\underset{\theta}{\arg\min} $ $ \mathcal{L}\left(\theta\right)
\quad \text{s.t. } \quad
h_1 \left(\theta \right)
= 0,  h_2 \left(\theta \right)
= 0, ...
$

where $\theta$ denotes the learnable parameters. When it comes to the case where $\theta$ represents the weights of a deep neural network, the constraints become extremely high-dimensional and nonlinear, especially when the number of constraints grows. Under this formulation, a constraint is only linear when $h(\theta)$ is a linear function of $\theta$. Therefore, the linearity does not hold when $\theta$ represents the weights of a neural network.

In our formulation, we only require the operator $\mathcal{F}$ to be linear, while the neural basis functions $\Psi$ can still be highly nonlinear with respect to $\theta$. Therefore, our formulation reduces the problem's complexity and allows us to explicitly determine and promote the existence of the solution.

### Comparison to soft constraint approaches

While there have been attempts to model constraints by overfitting an NN trained with regression, CNF has several clear advantages over such soft approaches:
- CNF satisfies hard constraints without training, while soft approaches may require extensive training.
- Despite extensive training, soft constraint approaches may fail to satisfy hard constraints due to inherent limitations in their learning capacity. In contrast, CNF provides a robust guarantee of hard constraint satisfaction within machine precision error, provided the condition number is small.
- Another drawback of employing soft approaches becomes evident when effectively imposing priors. The incorporation of priors often involves introducing an additional term to the loss. Consequently, a tradeoff arises between the constraints and the prior, which is controlled by a hyperparameter. In contrast, CNF offers a clean solution without such a tradeoff. With CNF, the inclusion of priors does not compromise hard constraints, thereby maintaining a harmonious balance among various aspects of the model.

### Generalization of NKF and PDE-CL

CNF generalizes the prior works NKF [38] and PDE-CL [25]. NKF employed dot-product kernel bases for 3D reconstruction, while PDE-CL used constraint bases to solve PDEs. However, their bases exhibit poor performance in terms of linear independence and learning capacity (refer to supplementary A2, B2). Additionally, their dense matrices make them unsuitable for large-scale problems. The simple dot-product kernel in NKF also cannot handle higher-order constraints (refer to supplementary A1 for explanations and C2 for evaluation).
We introduce several novel variations of basis functions to enhance linear independence, learning capacity, and matrix sparsity, along with strategies to analyze and promote solution existence. Our work unifies NKF and PDE-CL and demonstrates that CNF can be applied to general constrained optimization problems.

---

### Decision · Program_Chairs · 2023-09-21

**Decision:**

Accept (poster)

**Comment:**

The paper proposes a method for neural solvers for optimization problems with hard (linear) constraints. At a high level one can view this as a collocation method where the basis functions are learnable and parametrized by shallow neural networks. The net benefit is that the constraints are automatically enforced, and no tuning or regularization is necessary --- a property that is particularly helpful for enforcing higher order differential constraints while solving PDE systems.

The core idea is somewhat simple but well executed. The paper writing and evaluations across a range of applications are clearly communicated.  During the discussion phase, there were concerns raised about the methods' limitations (currently its scope is restricted to problems with linear constraints) but I feel that the authors responded well to these concerns. As such I am happy to recommend acceptance.